# Molecular Mechanisms of Cellular Senescence in Age-Related Endometrial Dysfunction

**DOI:** 10.3390/cells14120858

**Published:** 2025-06-06

**Authors:** Hiroshi Kobayashi, Mai Umetani, Miki Nishio, Hiroshi Shigetomi, Shogo Imanaka, Hiratsugu Hashimoto

**Affiliations:** 1Department of Gynecology and Reproductive Medicine, Ms.Clinic MayOne, 871-1 Shijo-cho, Kashihara 634-0813, Japan; mai_umetani@yahoo.co.jp (M.U.); mikin.yaruki10000.boshi@gmail.com (M.N.); shogo_0723@naramed-u.ac.jp (S.I.); hiratsugu_hashimoto@yahoo.co.jp (H.H.); 2Department of Obstetrics and Gynecology, Nara Medical University, 840 Shijo-cho, Kashihara 634-8522, Japan; hshige35@gmail.com; 3Department of Gynecology and Reproductive Medicine, Aska Ladies Clinic, 3-3-17 Kitatomigaoka-cho, Nara 634-0001, Japan

**Keywords:** AMPK, autophagy, endometrium, mTOR, p53, senescence

## Abstract

The endometrium is essential for reproductive function, supporting implantation and pregnancy through mechanisms such as hormonal responsiveness, immune regulation, and tissue regeneration. Aging disrupts these processes, with cellular senescence—marked by irreversible cell cycle arrest due to DNA damage and oxidative stress—being a key contributor. While senescence aids in tumor suppression and tissue repair, its dysregulation impairs endometrial function. Central to this regulation are p53, AMPK, and mTOR, which coordinate stress responses, metabolic regulation, and proliferation control. p53 activates AMPK and inhibits mTOR, promoting energy conservation and limiting senescence. AMPK also suppresses mTOR, reducing age-related dysfunction. This p53–AMPK–mTOR axis, along with autophagy, governs cell fate in response to stress and nutrient status. Although moderate senescence supports endometrial function, excessive accumulation can hinder fertility. Understanding these molecular interactions may advance fertility treatments and strategies to counteract reproductive aging.

## 1. Introduction

The endometrium is in direct contact with the external environment and is continuously exposed to various stimuli, such as infections and physical damage. To maintain homeostasis, it employs a multi-layered defense system, which includes protection against microbial invasion [1], remodeling in response to hormonal fluctuations, repair mechanisms against mechanical stress [2], and immune regulation [3]. These coordinated actions enable the endometrium to adapt to diverse external stimuli and fulfill its essential role in reproduction. Throughout a woman’s life, the endometrium undergoes significant physiological changes during key milestones such as menstruation, pregnancy, childbirth, and menopause. These changes are governed by a delicate balance between cellular regeneration and senescence, influenced primarily by hormonal activity. In particular, the regenerative capacity of the endometrium reflects the functional interplay between hormonal signals and endometrial cells and is critical for maintaining reproductive function [4]. Conversely, when cellular senescence associated with aging becomes dysregulated, it can impair endometrial receptivity and fertility and may contribute to conditions such as infertility, endometriosis, and endometrial cancer [4].

Cellular senescence is a tightly regulated physiological process triggered by factors such as DNA damage, oxidative stress, telomere shortening, oncogene activation, and nutritional stress [5,6]. Senescent cells exhibit several characteristic features, including irreversible cell cycle arrest; morphological alterations such as hypertrophy and cellular flattening [7]; increased activity of senescence-associated β-galactosidase (SA-β-gal); elevated expression of cell cycle inhibitors (p16^INK4a^ [*CDKN2A*] and p21^CIP1^ [*CDKN1A*]); and the accumulation of DNA damage markers such as γ-H2AX [7]. In addition, senescent cells acquire a senescence-associated secretory phenotype (SASP), characterized by the secretion of pro-inflammatory cytokines, chemokines, growth factors, and proteases [8], which can influence neighboring cells and the tissue microenvironment. The SASP contributes to wound healing, tissue repair, and tumor suppression [8,9,10], as well as tissue development during embryogenesis [8,11]. However, when senescent cells are not effectively cleared, their accumulation can result in chronic inflammation and promote the progression of age-related diseases such as atherosclerosis, osteoarthritis, and cancer [12].

The regulation of cellular senescence involves numerous genes and signaling networks associated with autophagy, the cell cycle, and metabolism [8]. Autophagy, a process that degrades and recycles unnecessary intracellular components, plays a pivotal role in maintaining cellular homeostasis and tissue remodeling. In the endometrium, cellular senescence is modulated by stress responses, inflammatory signaling, and intercellular communication, and it exerts a profound influence on endometrial function, embryo receptivity, and fertility [13,14]. Senescent cells also undergo significant metabolic alterations, including changes in mitochondrial function and energy metabolism [15]. Although signaling molecules such as the tumor suppressor p53, mechanistic target of rapamycin (mTOR), and AMP-activated protein kinase (AMPK) are known to be involved in cellular senescence, the mechanisms underlying the initiation of senescence and metabolic changes in the endometrium remain incompletely understood.

In this review, we outline the fundamental characteristics of autophagy and energy metabolism associated with cellular senescence, examine the roles and interactions of p53, mTOR, and AMPK, present the latest findings on endometrial cellular senescence, and discuss potential therapeutic strategies for addressing age-related declines in reproductive function.

## 2. Methods

### Search Strategy and Selection Criteria

In this narrative review, we performed an extensive literature survey centered on autophagy and energy homeostasis in the context of endometrial cellular senescence, with particular emphasis on the roles of p53, AMPK, and mTOR. Relevant studies published up to 31 March 2025 were identified through searches in two electronic databases: PubMed (https://pubmed.ncbi.nlm.nih.gov/ accessed on 19 May 2025) and Google Scholar (https://scholar.google.com/ accessed on 19 May 2025), utilizing a combination of the following keywords: “endometrium”, “reproduction”, “cellular senescence”, “autophagy”, “p53”, “AMPK”, and “mTOR”. This review does not constitute a systematic review employing a predefined research framewo rk but rather seeks to provide a comprehensive and integrative overview of the current state of knowledge on this subject.

## 3. Factors Involved in Endometrial Cellular Senescence

Endometrial senescence is regulated by the interplay of multiple molecular mechanisms, including inflammatory cytokines, intracellular signaling pathway regulators, and cell cycle modulators. Among these, the SASP plays a central role in both the progression of cellular senescence and its effects on surrounding tissues. Inflammatory cytokines such as interleukin-17 receptor B (IL17RB) and interleukin-1β (IL-1β) activate the c-Jun N-terminal kinase (JNK) pathway, thereby promoting senescence in endometrial stromal cells and stimulating the secretion of SASP factors [16,17]. Plasminogen activator inhibitor-1 (PAI-1), secreted by senescent stromal cells, acts as a master regulator of paracrine senescence signals, amplifying SASP expression in neighboring cells [18]. Moreover, insulin-like growth factor-binding protein 3 (IGFBP3), which is upregulated in aged mesenchymal stem cells, has been shown to be regulated by the PI3K (phosphoinositide 3-kinase)/Akt (protein kinase B) pathway and to contribute to SASP-independent paracrine senescence [19]. These inflammatory mediators play critical roles in the progression of the chronic inflammatory state characteristic of aging tissues, commonly referred to as “inflammaging” [19]. Inflammaging, characterized by chronic, low-grade inflammation linked to aging, plays a pivotal role in promoting endometrial senescence, thereby contributing to endometrial dysfunction [20]. This process involves the secretion of pro-inflammatory mediators collectively known as the SASP, and the resulting sustained inflammatory milieu disrupts tissue homeostasis and diminishes endometrial receptivity, ultimately impairing fertility and pregnancy outcomes [20]. To date, several comprehensive review articles have explored the relationship between cellular senescence and inflammaging. Saito et al. discuss the dynamic interplay between SASP-mediated cellular senescence and inflammaging [21], while Lee et al. use the skin as a model to study these phenomena [22]. Other studies have examined the involvement of cellular senescence and inflammaging in traumatic brain injury [23] and their impact on bone health [24]. Li et al. provide a broad overview of cellular senescence, its role in aging and disease, and potential therapeutic strategies [25]. Collectively, these studies offer important insights into how cellular senescence and inflammaging contribute to aging and age-related pathologies.

Additionally, Cbl proto-oncogene-like 1 (CBLL1) and cell division cycle 42 (CDC42) have emerged as significant regulators of signaling pathways involved in senescence. Elevated CBLL1 expression in senescent endometrial stromal cells influences cellular senescence and proliferation by downregulating the tumor suppressor phosphatase and tensin homolog (PTEN) and increasing intracellular reactive oxygen species (ROS) levels [26]. Furthermore, the upregulation of CDC42—a member of the Rho GTPase family—induces premature senescence in stromal cells and is associated with reduced endometrial receptivity and implantation capacity [27]. Umbayev et al. have discussed the involvement of CDC42 in cellular senescence and its potential as a therapeutic target for age-related diseases [28]. Moreover, Ito et al. demonstrated that CDC42 mediates chronic inflammation associated with endothelial senescence and contributes to atherosclerosis development [29].

The tumor suppressor p53 and the signaling molecules AMPK and mTOR are also key regulators of cellular senescence. To better understand their functional roles in controlling senescence, current research has focused on several key areas. The identification of autophagy machinery components and metabolism-related determinants that influence whether p53 activation leads to cellular senescence or reversible quiescence, as well as the elucidation of the underlying mechanisms. Additionally, a detailed investigation of the context-dependent role of AMPK in senescence, particularly its crosstalk with the p53 and mTOR signaling pathways under various cellular stress conditions. Furthermore, a deeper understanding of the molecular mechanisms through which mTOR activity induces and sustains senescence-associated phenotypes remains a pressing challenge.

In the following section, we individually examine the molecular mechanisms by which major senescence-associated genes—including p53, mTOR, and AMPK—regulate endometrial cell senescence.

### 3.1. p53

The p53 protein, encoded by the TP53 gene located on chromosome 17p13.1, responds to a wide range of endogenous and exogenous stress stimuli—such as DNA damage, oxidative stress, and oncogene activation—and plays a critical role in maintaining genomic stability by arresting the cell cycle and inducing the transcription of DNA repair-related genes [30]. p53 is widely recognized as a tumor suppressor [31] and functions as a central transcription factor that induces cellular senescence, an irreversible form of cell cycle arrest [32,33,34]. In cases where DNA damage is irreparable, p53 activates the apoptotic pathway to eliminate damaged cells.

#### 3.1.1. Functional Role of p53 in Cellular Senescence and Autophagy Regulation

The activity of p53 is tightly regulated by negative modulators, such as the E3 ubiquitin ligase mouse double minute 2 homolog (MDM2), which suppresses p53 protein accumulation and transcriptional activity to prevent overactivation [35]. Activation of p53 has been reported to regulate SASP expression via mTOR inhibition, thereby suppressing the secretion of inflammatory cytokines [36,37,38] (Figure 1 ➀). In addition, p53 halts the cell cycle at the G1/S checkpoint through cyclin-dependent kinase inhibitors, allowing DNA damage to be repaired prior to cell division [39,40]. It also supports genome integrity by promoting the transcription of DNA repair factors such as damage-specific DNA-binding protein 2 (DDB2) and xeroderma pigmentosum group C (XPC) [41]. When DNA damage is reversible, p53-mediated senescence induction is circumvented (Figure 1 ➁). Conversely, in the presence of irreparable DNA damage, p53 activates the transcription of pro-apoptotic genes such as BAX (BCL2-associated X, apoptosis regulator), PUMA (p53 upregulated modulator of apoptosis), and PMAIP1 (phorbol-12-myristate-13-acetate-induced protein 1), triggering cell death and preventing the accumulation of abnormal cells [36,42] (Figure 1 ➂). Thus, as a master regulator of the cellular stress response, p53 orchestrates diverse cell fate decisions—ranging from DNA repair and cell cycle control to apoptosis and senescence—thereby preserving cellular and tissue homeostasis and suppressing tumorigenesis [30]. Furthermore, p53 plays a crucial role in metabolic regulation. It suppresses glycolysis [43], promotes oxidative phosphorylation (OXPHOS) [44], and induces the expression of antioxidant enzymes such as superoxide dismutase (SOD) and catalase. It also regulates glutathione metabolism and maintains mitochondrial function, thereby contributing to the suppression of cellular senescence [45] (Figure 1 ➃). Consequently, p53 has been suggested to possess a context-dependent dual role—either promoting or suppressing senescence depending on the cellular environment [32,46]. Excessive senescence may lead to tissue dysfunction and age-related diseases, whereas inadequate senescence responses may permit abnormal cell proliferation and tumorigenesis.

Additionally, autophagy plays an important role in regulating cellular senescence. This intracellular homeostatic mechanism is activated by various stressors, including nutrient deprivation, hypoxia, DNA damage, and cytotoxicity. Autophagy maintains mitochondrial and overall cellular function by degrading and recycling organelles and abnormal proteins via a lysosome-dependent pathway [47,48], thus mitigating the accumulation of intracellular damage and slowing the progression of cellular senescence. The key regulators of autophagy include p53, AMPK, and mTOR. p53 is known to inhibit mTOR signaling through both transcription-dependent and -independent mechanisms [32,49] (Figure 1 ➄). As a transcription factor, p53 induces the expression of several genes—such as REDD1 (regulated in development and DNA damage responses 1), TSC2 (TSC complex subunit 2), PTEN, LKB1 (liver kinase B1), the AMPKβ subunit, and Sestrin1/2—which act directly or indirectly to suppress mTORC1 (mechanistic Target of Rapamycin Complex 1) activity [50,51,52]. Moreover, p53 has been shown to directly bind the promoters of autophagy-related genes (ATGs), thereby positively regulating autophagy [32]. In terms of transcription-independent mechanisms, p53 suppresses the translation of mTOR pathway components (e.g., mTOR, ribosomal S6 kinase (S6K1), and Rictor) by modulating the expression of microRNAs such as miR-34, miR-145, and miR-155 [32]. AMPK, activated by intracellular adenosine 5’ triphosphate (ATP) depletion or increased AMP levels, phosphorylates TSC2 to enhance its GTPase-activating protein (GAP) activity, thereby inactivating Rheb (Ras homolog, mTORC1 binding) and inhibiting mTORC1. Through this molecular network, autophagy contributes to the regulation of energy metabolism and the suppression of cellular senescence [53] (Figure 1 ➅). In addition, the p53 family members p63 and p73 also play important roles in maintaining cellular metabolic homeostasis. Berkers et al. demonstrated that the p53 family mediates the link between energy stress responses and fate-determining processes such as cellular senescence by activating AMPK and suppressing mTORC1 [54]. The p53 family is involved in a variety of metabolic processes, including regulation of the AMPK/mTOR pathway, glucose and lipid metabolism, mitochondrial function, redox balance, and autophagy. Cellular senescence suppresses cell growth and proliferation through p53 activation [52] (Figure 1 ➆). Meanwhile, amino acids and metabolites generated by autophagy can activate mTORC1 within intracellular structures called TASCCs (TOR-Autophagy Spatial Coupling Compartments), thereby promoting the mass production of SASP factors [55] (Figure 1 ➆). The formation of TASCCs spatially integrates the catabolic activity of autophagy with the anabolic signaling of mTORC1, enhancing SASP production in senescent cells. These findings suggest that autophagy may also contribute to the promotion of cellular senescence in a context-dependent manner [36,56].

Furthermore, p53 is known to both promote and suppress autophagy depending on its subcellular localization. Nuclear p53 acts as an autophagy activator by enhancing the transcription of autophagy-inducing genes such as DNA damage-regulated autophagy modulator 1 (DRAM1), and by regulating histone-modifying enzymes to promote the transcription of ATG genes [57] (Figure 1 ➇). Conversely, cytoplasmic p53 has been reported to suppress autophagy [58], highlighting the dual and complex role of p53 in autophagy regulation (Figure 1 ➈). Additionally, certain mutant forms of p53 can suppress autophagy via activation of the mTOR pathway [59] (Figure 1 ➉). Targeting these mutant forms may alleviate autophagy suppression and restore its functional activation. Collectively, these findings indicate that autophagy is a condition-dependent process that may either promote or suppress cellular senescence. Its biological effects vary significantly depending on factors such as the subcellular localization of p53, mutation status, cellular environment, and stress conditions [60].

#### 3.1.2. Functional Role of p53 in Metabolism

Cellular senescence differs from the reversible quiescent state (G_0_ phase), in which cells temporarily cease division in response to external stresses such as nutrient deprivation or growth factor withdrawal, and from terminal differentiation, in which nerve and muscle cells permanently exit the cell cycle and acquire specialized functions [8]. Senescent cells permanently lose their proliferative capacity due to various stress responses—such as DNA damage, oxidative stress, and telomere shortening—but remain metabolically active and undergo distinctive metabolic reprogramming [8]. Senescent cells often exhibit increased glycolytic flux and enhanced lactate production [8]. In addition, significant changes occur in mitochondrial function and lipid metabolism, including impaired mitochondrial function despite an increase in mitochondrial mass [61,62]. Reports on fatty acid β-oxidation are inconsistent, with some studies showing an increase [63] and others a decrease [62], suggesting that metabolic imbalance may lead to intracellular lipid droplet accumulation. These metabolic adaptations are thought to compensate for the energy demands associated with sustaining the SASP [8].

Below, we outline the metabolic regulatory mechanisms of p53, a central transcription factor in the progression of cellular senescence (Figure 2, left). p53 modulates energy metabolism primarily by suppressing glycolysis and enhancing mitochondrial OXPHOS [43,64,65,66]. Specifically, p53 represses the expression of glucose transporters GLUT1 (glucose transporter type 1), GLUT3, and GLUT4, thereby limiting glucose uptake and reducing substrate availability for glycolysis [65,67]. Moreover, p53 induces the expression of TP53-induced glycolysis and apoptosis regulator (TIGAR), which lowers the activity of phosphofructokinase-1 (PFK1), a key glycolytic enzyme [45]. As a result, fructose-6-phosphate (F6P) accumulates in the cell. Under oxidative stress, the increased consumption of nicotinamide adenine dinucleotide phosphate (NADPH) raises the NADP^+^/NADPH ratio, enhancing the activity of glucose-6-phosphate dehydrogenase (G6PDH) and diverting glucose metabolism from glycolysis to the pentose phosphate pathway (PPP). The PPP provides ribose-5-phosphate for nucleotide synthesis, supporting the biosynthetic demands of senescent cells. Additionally, increased NADPH production through the PPP enhances antioxidant capacity and protects against oxidative damage. However, excessive PPP activity can disturb metabolic homeostasis, and p53 counteracts this by directly inhibiting G6PDH, the rate-limiting enzyme of the PPP [68].

p53 also suppresses the activity of phosphoglycerate mutase (PGM), thereby inhibiting the conversion of glucose to pyruvate [69]. Pyruvate metabolism is regulated by pyruvate dehydrogenase (PDH) and its inhibitor, pyruvate dehydrogenase kinase (PDK). PDH converts pyruvate to acetyl-CoA, linking glycolysis to the tricarboxylic acid (TCA) cycle and OXPHOS [70]. In contrast, PDK phosphorylates and inactivates PDH, redirecting pyruvate toward lactate production via lactate dehydrogenase (LDH) [70]. The activity of the PDK/PDH axis varies among senescent cells depending on the type and cause of senescence. For example, in senescent human prostate stromal cells, upregulation of PDK4 inhibits PDH and promotes aerobic glycolysis (the Warburg effect), increasing glucose uptake and lactate production [71]. In contrast, in cancer cells undergoing senescence induced by the BRAF^V600E^ mutation, PDK1 expression is suppressed, leading to enhanced oxidative metabolism via the TCA cycle [70,72]. These findings highlight the metabolic flexibility of senescent cells, which remodel their pathways in response to environmental and intracellular cues to balance glycolysis and OXPHOS according to changing energy and biosynthetic needs [73].

Furthermore, p53 promotes energy-efficient metabolic pathways to maintain energy homeostasis under stress. It induces the expression of Lipin1, a regulator of fatty acid β-oxidation [74], and GAMT (guanidinoacetate methyltransferase), which participates in the creatine-phosphocreatine shuttle, an alternative system for ATP production [75]. These adaptive metabolic changes support cell survival, particularly under nutrient-limited conditions.

In addition, p53 promotes OXPHOS by activating the transcription of synthesis of cytochrome c oxidase 2 (SCO2), a gene essential for the assembly of complex IV, thereby ensuring efficient mitochondrial ATP production [76]. When SCO2 expression is reduced due to p53 deficiency, mitochondrial respiration becomes impaired, leading to a compensatory shift toward glycolysis [76]. Similarly, p53 induces the expression of the mitochondrial enzyme glutaminase 2 (GLS2), which catalyzes the conversion of glutamine to glutamate and subsequently to α-ketoglutarate (α-KG), a key intermediate in the TCA cycle. This pathway not only supports oxidative metabolism but also contributes to the antioxidant defense mechanism [77,78]. Apoptosis-inducing factor (AIF), while known for its role in promoting cell death, is also essential for maintaining mitochondrial function. p53 may enhance AIF expression, thereby stabilizing mitochondrial complex I and improving the efficiency of OXPHOS, which in turn suppresses excessive ROS production and prevents oxidative damage [79]. Additionally, the p53-inducible ribonucleotide reductase subunit p53R2 (RRM2B) plays a crucial role in deoxyribonucleotide synthesis and mitochondrial DNA (mtDNA) replication, supporting mitochondrial function and sustaining energy production under stress conditions [80]. p53 also activates the transcription of Parkin (PARK2), an E3 ubiquitin ligase, thereby promoting mitochondrial respiration and inhibiting glycolysis [81]. Notably, nuclear-localized p53 enhances Parkin transcription, whereas cytoplasmic p53 inhibits Parkin activity in mitophagy [82]. Through Parkin, p53 contributes to the regulation of mitochondrial quality control and cellular metabolism.

Therefore, p53 activation in response to DNA damage under metabolic stress often induces cellular senescence, reflecting its role as an energy sensor. In general, p53 functions to suppress glycolysis and enhance mitochondrial metabolism [64]. However, exceptions exist—for instance, in pancreatic β-cells and hepatocytes, p53 has been reported to promote glycolysis and inhibit OXPHOS [83]. These observations underscore the context-dependent nature of p53-mediated metabolic regulation, which varies with cell type and physiological or pathological conditions. Moreover, p53 regulates the oxidative stress response and influences cell lifespan by modulating intracellular ROS levels. Through its integrated roles in metabolic homeostasis and cellular senescence, p53 acts as a key regulator of cellular function and a critical suppressor of tumorigenesis [80]. In addition, it has been suggested that p53 may contribute to the maintenance of reproductive function by regulating autophagy, cellular senescence, energy metabolism, and redox balance.

#### 3.1.3. Functional Role of p53 in Maintaining Reproductive Function

In the endometrium, the role of p53 in regulating autophagy and cellular senescence has been increasingly clarified based on recent findings. In normal endometrial tissue, autophagy is a fundamental physiological process required for maintaining cellular homeostasis, tissue remodeling, and adaptation to fluctuations in the hormonal environment. Estrogen, in particular, regulates the transcription of autophagy-related genes through its receptors (ERα, ERβ, and GPER (G protein-coupled estrogen receptor 1)). Moreover, estrogen modulates autophagy activity through multiple layers of epigenetic and post-transcriptional mechanisms, including the action of transcription factors, microRNAs, and histone modifications [84]. Endometrial cells dynamically modulate autophagy activity in response to cyclic changes in estrogen and progesterone concentrations during the menstrual cycle. This regulation enables the functional restructuring of the endometrium in preparation for embryo implantation and pregnancy [47]. In particular, the decline in ovarian steroid hormone levels during the late menstrual phase is believed to activate autophagy, contributing to endometrial breakdown, clearance of cellular debris, and immune system activation (Figure 1, bottom left). Autophagy activity markedly increases during the transition from the secretory phase to the menstrual phase [84], underscoring its importance in the cyclic regeneration of the endometrium. Conversely, impaired or dysregulated autophagy can result in defective decidualization and severely compromise endometrial function [85]. Abnormalities in autophagy have also been associated with pathological conditions such as infertility [86], endometriosis [87], and endometrial cancer [88]. Furthermore, estrogen deficiency has been shown to enhance autophagy and induce apoptosis in endometrial epithelial cells in both pre- and postmenopausal women [89] and in ovariectomized rat models [90]. These findings suggest that the coordinated regulation of autophagy and apoptosis plays a role in endometrial atrophy. Age-related declines in estrogen levels are thought to alter autophagy activity, contributing to reduced endometrial function. Thus, ovarian steroid hormones are critically involved in maintaining the homeostasis of reproductive tissues through the regulation of autophagy [89,90].

In parallel with these shifts in autophagy, p53 expression in the endometrium is also known to vary in response to hormonal cues. Estrogen has been reported to enhance p53 promoter activity via transcription factors such as Nuclear Factor kappa B (NF-κB) and Sp1, leading to increased transcription of the p53 gene [91]. Notably, p53 expression rises significantly during the proliferative phase of the menstrual cycle, suggesting a role for p53 in cell cycle regulation and DNA damage repair during periods of active cell proliferation and tissue remodeling [92]. These findings imply that hormone-dependent regulation of p53 is critical for maintaining endometrial functional integrity. Moreover, studies using a uterus-specific p53 knockout mouse model have demonstrated reduced decidual development and increased cellular senescence, supporting the notion that p53 is essential for endometrial functional maturation and the regulation of senescence [93]. However, the reported dynamics of p53 expression across the menstrual cycle are inconsistent. For example, one study reported higher p53 expression during the secretory phase compared to the proliferative phase [94]. Therefore, further research is needed to clarify the relationship between hormonal fluctuations and p53 expression in the endometrium.

p53 also plays a critical role in maintaining reproductive function. In p53-deficient female mice, the expression of leukemia inhibitory factor (LIF)—a cytokine essential for embryo implantation—was reduced, resulting in significantly lower implantation and pregnancy rates. These findings suggest that p53 contributes to successful implantation by promoting the transcription of LIF [95]. Feng et al. demonstrated that the p53 family (p53, p63, and p73) plays a central role in female reproductive function, particularly in embryo implantation and pregnancy maintenance [96]. In p63-deficient mice, diminished ovarian function and reduced fertility were observed, indicating that p63 is involved in the formation of mature oocytes and the regulation of apoptosis within the ovary. In p73-deficient mice, increased spindle abnormalities and chromosomal aneuploidy were found in oocytes, suggesting that p73 contributes to the spindle assembly checkpoint that ensures accurate chromosome segregation during meiosis. Furthermore, in humans, a specific single nucleotide polymorphism (SNP) in the p73 gene was found to be significantly more prevalent among in vitro fertilization (IVF) patients aged 35 years or older, suggesting a link to poor oocyte quality and implantation failure [96]. Collectively, these findings indicate that the p53 family acts as a key regulator of reproductive processes, including embryo implantation and the preservation of oocyte quality. In addition, a study by Delenko et al. reported that quercetin—a flavonoid with antioxidant and senolytic properties—promotes the decidualization of human endometrial stromal cells via the AKT–ERK–p53 signaling pathway [97]. Their findings suggest that quercetin may enhance decidualization in patients with endometriosis and thereby contribute to improved endometrial function. Experimental studies further support the essential role of p53 in decidualization, showing that p53 activation promotes this process while its inhibition suppresses it. Taken together, these studies strongly suggest that p53 is intricately involved in the regulation of cellular senescence, autophagy, energy metabolism, and decidualization within the endometrium and that it plays a pivotal role in maintaining female reproductive function.

### 3.2. AMPK

AMPK is a crucial metabolic sensor that maintains cellular energy homeostasis by responding to fluctuations in intracellular energy status. It is activated under energy stress conditions such as glucose deprivation, reduced ATP levels, and increased concentrations of AMP and ADP [98]. To restore cellular energy balance, AMPK promotes glucose uptake and fatty acid β-oxidation and facilitates ATP resynthesis while simultaneously inhibiting ATP-consuming anabolic pathways, including lipid and protein synthesis. Additionally, AMPK contributes to metabolic reprogramming by inducing mitochondrial biogenesis and autophagy, thereby supporting cellular adaptation and survival under stress conditions. AMPK activation is mediated by several upstream kinases, primarily LKB1 [99], CaMKKβ (calcium/calmodulin-dependent protein kinase kinase β) [99], and ATM (ataxia telangiectasia mutated) [100]. LKB1 phosphorylates and activates AMPK in response to energy depletion, such as decreased ATP or elevated AMP levels. CaMKKβ is involved in AMPK activation in response to increased intracellular Ca^2+^ concentrations ([Ca^2+^]_i_), while ATM, a serine/threonine kinase primarily involved in the DNA damage response, activates AMPK in the context of oxidative stress and DNA damage-induced cellular stress. These kinases dynamically regulate AMPK activity in response to diverse stress signals, playing a key role in preserving metabolic homeostasis and promoting cell survival. Moreover, AMPK modulates the nutrient-sensitive sensor mTORC1, either directly or indirectly, thereby inhibiting anabolic processes and promoting catabolic pathways to reestablish metabolic balance [101,102] (see Figure 1 ➄). AMPK also induces autophagy while preventing its excessive activation, contributing to the proper regulation and maintenance of the autophagic machinery and overall cellular homeostasis [102]. Thus, AMPK functions not only as an energy sensor but also as a central regulator of autophagy, making it indispensable for stress adaptation and cellular equilibrium. Additionally, AMPK is thought to regulate autophagy through crosstalk with the p53 and mTOR pathways and to play a significant role in reproductive cell functions, including cellular senescence and the decidualization of endometrial stromal cells.

#### 3.2.1. Functional Roles of AMPK in Cellular Senescence and Autophagy Regulation

For detailed information regarding the signaling pathways involving upstream regulators and downstream targets of AMPK, please refer to previous studies [98,102]. This subsection focuses on the dual role of AMPK in the regulation of cellular senescence. When intracellular energy levels decline, AMPK becomes activated and regulates the activity of mTORC1 via downstream targets such as TSC2 and Rheb [103] (Figure 3 ➀). mTORC1 is a key regulator of cell growth and protein synthesis, and its suppression contributes to energy conservation and metabolic reprogramming. Under energy-deprived conditions such as glucose starvation and ATP depletion, AMPK activation leads to the phosphorylation of ULK1 (Unc-51-Like Autophagy-Activating Kinase 1), inducing autophagy. This facilitates the selective removal of damaged organelles and misfolded proteins, thereby contributing to the maintenance of cellular homeostasis and the suppression of aging [104,105] (Figure 3 ➁). Conversely, recent studies have suggested that AMPK may also negatively regulate ULK1 activity, thereby suppressing autophagy [106,107]. It has also been reported that AMPK can differentially regulate both the early (autophagosome formation) and late (lysosomal activation) phases of autophagy through distinct mechanisms [107]. These findings suggest that AMPK may either promote or inhibit autophagy via ULK1, depending on the cell’s energy status, nutrient availability, and type of stress, indicating a dynamic and context-dependent interaction. In this way, the AMPK–ULK1 interaction appears to play a “dual” role, maintaining basal autophagic function while preventing its excessive activation [102]. Moreover, AMPK enhances the activity of nicotinamide phosphoribosyltransferase (NAMPT), thereby promoting the biosynthesis of NAD^+^ (oxidized nicotinamide adenine dinucleotide), which serves a protective role against oxidative stress-induced cellular senescence [108,109] (Figure 3 ➂). Elevated NAD^+^ levels enhance the activity of the NAD^+^-dependent deacetylase SIRT1 (sirtuin 1), which in turn induces the deacetylation and transcriptional activation of PGC-1α (peroxisome proliferator-activated receptor gamma coactivator 1-alpha) and the FOXO (Forkhead box O) transcription factors [110]. This pathway promotes mitochondrial biogenesis and function, contributing significantly to the extension of cellular lifespan and enhancement of stress resistance. Under acute stress or energy-deprived conditions, AMPK also suppresses inflammation by inhibiting the NF-κB (Nuclear Factor kappa B)–SASP–STAT3 (Signal Transducer and Activator of Transcription 3) signaling axis, thereby mitigating the progression of aging [111,112] (Figure 3 ➃). In many cell types, AMPK activation has demonstrated anti-aging effects through the maintenance of metabolic homeostasis and enhancement of oxidative stress resistance.

On the other hand, under metabolic stress conditions such as glucose deprivation, AMPK activates p53 (tumor protein p53), leading to cell cycle arrest [113,114] (Figure 3 ➄). Activated p53 induces the transcription of the cyclin-dependent kinase inhibitor p21, triggering permanent cell cycle arrest (senescence) or apoptosis [113,115] (Figure 3 ➅). Mathematical modeling studies by Zhou and Liu have proposed a mechanism in which the AMPK–p53 axis governs cell cycle arrest: if glucose levels recover early, cells can resume proliferation, but if energy depletion persists, moderate p53 activation sustains senescence, and prolonged stress may lead to high p53 expression and inhibition of mTOR and Akt, resulting in apoptosis [113]. The biphasic activation of p53 acts as a critical switch in cell fate determination. The mTOR pathway may further regulate p53 activity either positively or negatively, depending on cell type and stress context [32]. Thus, under excessive metabolic stress, AMPK may also promote aging. In the DNA damage response, ATM kinase phosphorylates p53 and induces downstream expression of p21 [116] (Figure 3 ➆). AMPK can also be activated by ATM [117], potentially contributing to the induction of cellular senescence (Figure 3 ➇). Furthermore, a reduced intracellular NAD^+^/NADH ratio can activate AMPK via ATP depletion, which in turn promotes p53 phosphorylation and accelerates senescence [118] (Figure 3 ➈→➄→➅). Interestingly, in human dental follicle cells, both excessive activation and suppression of AMPK have been reported to induce cellular senescence [119]. Additionally, long-term administration of AMPK activators such as metformin or AICAR (5-aminoimidazole-4-carboxamide ribonucleoside) may promote aging through autophagy dysfunction and mitochondrial abnormalities [120]. Meanwhile, p53 also has anti-inflammatory effects, such as suppressing the expression of inflammatory cytokines like IL-1 [121]. Therefore, while the AMPK–p53 axis contributes to the establishment of senescence via growth arrest, it also suppresses inflammatory components of the SASP, serving a dual function. This may lead to a less inflammatory phenotype—referred to as “silent senescence”—which could help prevent the progression of diseases driven by chronic inflammation [121].

In summary, under conditions of energy stress or oxidative damage, AMPK functions to suppress cellular senescence by regulating autophagy, maintaining NAD^+^ homeostasis, and inhibiting inflammatory responses. However, depending on the duration of its activation, cell type, and nature of the stimuli, AMPK can also promote senescence by inducing mitochondrial dysfunction or the expression of senescence-associated genes. Thus, AMPK is emerging as a dual-faced and context-dependent modulator of cellular senescence.

#### 3.2.2. Functional Role of AMPK in Metabolism

AMPK functions as a central sensor for maintaining intracellular energy homeostasis, and its activation is closely linked to cellular metabolic adaptation. Upon activation, AMPK orchestrates multiple metabolic pathways, particularly by promoting ATP-generating processes such as glycolysis, thereby eliciting a metabolic response to energy deficiency [122,123]. Specifically, AMPK suppresses gluconeogenesis by downregulating the expression of gluconeogenic enzymes such as phosphoenolpyruvate carboxykinase (PEPCK) and glucose-6-phosphatase (G6Pase) [124,125] while simultaneously enhancing the translocation of the glucose transporter GLUT4 to the plasma membrane, thereby increasing glucose uptake into the cell [126,127]. In addition, AMPK activation enhances the activity of key glycolytic enzymes such as hexokinase, PFK1, and PFK2, which rapidly restore intracellular ATP levels [127,128]. Concurrently, AMPK inhibits anabolic pathways to conserve energy. For instance, it phosphorylates and inactivates 3-hydroxy-3-methylglutaryl-CoA reductase (HMG-CoA reductase), a rate-limiting enzyme in the cholesterol biosynthesis pathway, thereby suppressing the mevalonate pathway and downstream cholesterol synthesis, ultimately reducing ATP consumption [129]. Similarly, AMPK inhibits acetyl-CoA carboxylase (ACC), a key enzyme in fatty acid synthesis, through phosphorylation, contributing further to the conservation of cellular energy [126,127]. Conversely, under conditions of sufficient energy availability or in scenarios requiring long-term metabolic restructuring, AMPK suppresses glycolysis and promotes a metabolic shift toward oxidative pathways. This shift involves enhanced mitochondrial β-oxidation and the activation of transcriptional coactivators such as PGC-1α, which facilitate mitochondrial biogenesis and enhance oxidative phosphorylation [126,130]. Through this mechanism, AMPK enables cells to transition from glycolysis to the more efficient ATP production pathway of oxidative phosphorylation. Moreover, AMPK activation strongly supports oxidative phosphorylation by modulating transcription factors, phosphorylating substrate proteins, regulating mitochondrial dynamics, and inducing autophagy, particularly selective mitochondrial degradation (mitophagy) [130,131]. Through these actions, AMPK serves as a master regulator of metabolism, promoting ATP-generating pathways while suppressing energy-consuming anabolic processes to maintain energy homeostasis. Notably, AMPK exhibits bidirectional regulation of glycolysis, promoting or suppressing it depending on the metabolic context. This dual capability underpins its role as a “metabolic master switch,” allowing cells to swiftly adapt to acute energy deprivation while also facilitating long-term metabolic optimization and homeostatic maintenance. Through this flexible regulatory capacity, AMPK plays a pivotal role in ensuring cellular survival and functional integrity under various environmental changes and stress conditions.

#### 3.2.3. Functional Role of AMPK in the Maintenance of Reproductive Function

AMPK plays a central role in suppressing the progression of cellular senescence through the regulation of autophagy and intracellular metabolic homeostasis. In skeletal muscle, cardiac muscle, and the central nervous system, the responsiveness to AMPK activation has been reported to decline markedly with age [98]. One proposed mechanism for this phenomenon is that age-related chronic inflammation—referred to as “inflammaging”—may inhibit AMPK activation [98]. Inducers of cellular senescence include oxidative stress, chronic inflammation, aberrant accumulation of [Ca^2+^]_i_, and the activation of DNA damage response pathways. These factors are associated with the aging of endometrial stem cells and defective decidualization [132]. In particular, [Ca^2+^]_i_ is known to be involved in cell fate determination processes such as autophagy, apoptosis, and necrosis under oxidative stress conditions, and its precise regulation is critically important for the suppression of cellular senescence [133]. The removal of intracellular calcium at high concentrations has been shown to induce AMPK-dependent autophagy and may suppress premature senescence in human endometrial stromal cells [133].

Several studies have demonstrated that AMPK plays an important role in female reproductive function, particularly in maintaining metabolic homeostasis, suppressing cellular senescence, and regulating decidualization. For example, Kawano et al. reported that inflammatory cytokine IL-1β enhances the production of IL-8, MCP-1 (monocyte chemoattractant protein-1), PGE_2_ (prostaglandin E_2_), and PGF_2_α(prostaglandin F_2_α) in endometrial stromal cells. However, treatment with the AMPK activator AICAR significantly reduced the production of these inflammatory mediators [134]. This finding suggests that AMPK activation may mitigate excessive inflammatory responses in the endometrium. Furthermore, experiments using genetically modified mice lacking the catalytic AMPK subunits Prkaa1 and Prkaa2, specifically in the uterus (Prkaa1/2 d/d mice), revealed impaired endometrial regeneration postpartum, accompanied by fibrosis and structural disorganization [135]. Notably, AMPK inhibited hormonally induced artificial decidualization without affecting ovarian function or hormone production, indicating that AMPK is indispensable for decidualization and uterine tissue repair after parturition [135]. Another study demonstrated that AMPK deficiency impairs the proliferation of endometrial epithelial cells and disrupts the decidualization process, suggesting a crucial role for AMPK in hormonal responsiveness, endometrial receptivity, and embryo implantation during early pregnancy [136]. A study by Zhang et al. further reported that folate deficiency suppresses autophagy via the AMPK/mTOR signaling pathway, thereby impairing the decidualization process [137]. This finding supports the notion that AMPK contributes to the functional differentiation of the endometrium via autophagy, playing a critical role in the establishment of pregnancy.

Collectively, these findings clearly demonstrate that AMPK functions as a key regulatory factor in hormonal response modulation, endometrial regeneration, decidualization, and overall female reproductive function. Therefore, AMPK dysfunction is likely to contribute to the pathogenesis of infertility and pregnancy-related complications. Elucidating the molecular mechanisms underlying AMPK activity may provide important insights into the development of novel therapeutic targets in reproductive medicine.

### 3.3. mTOR

mTOR is classified as a serine/threonine-specific protein kinase. It functions as a central intracellular signaling molecule that integratively regulates diverse cellular processes—such as protein synthesis, lipid metabolism, and autophagy—in response to extracellular and intracellular nutritional status, energy levels, and the presence of growth factors [138]. As a “master regulator” of cellular growth, metabolism, proliferation, and survival, mTOR forms two functionally distinct multi-protein complexes: mTORC1 and mTORC2, each of which governs specific biological processes in response to different stimuli [32,113,139]. mTORC1 is activated by growth factors such as insulin and insulin-like growth factors (IGFs), essential amino acids (especially leucine and arginine), and high-energy states (e.g., elevated intracellular ATP levels). Once activated, mTORC1 phosphorylates ribosomal S6K1 and eIF4E-binding protein 1 (4E-BP1), thereby promoting mRNA translation and inducing protein synthesis [139,140]. Additionally, mTORC1 promotes lipid biosynthesis by regulating the expression and activity of enzymes involved in lipid synthesis, and it activates de novo purine and pyrimidine nucleotide synthesis pathways, supporting the supply of biosynthetic substrates essential for cell proliferation. In this way, mTORC1 contributes to cell growth and homeostasis by promoting anabolic pathways and simultaneously suppressing catabolic processes such as autophagy. On the other hand, mTORC2 is primarily activated by growth factor stimulation and T-cell receptor signaling. It induces the activation of Akt through phosphorylation at Ser473, thereby regulating processes including cell survival, energy metabolism, and actin cytoskeleton remodeling [139]. Moreover, mTORC2 plays a critical role in insulin signaling and is also involved in metabolic regulation, such as lipogenesis [141].

#### 3.3.1. Functional Role of mTOR in Cellular Senescence and Autophagy Regulation

This subsection focuses on the functional significance of mTOR in the regulation of autophagy and cellular senescence. The PI3K/AKT/mTOR pathway is a pivotal intracellular signaling cascade that orchestrates a wide spectrum of cellular functions, including growth, proliferation, metabolism, and survival. Upon activation by growth factors and other extracellular cues, the PI3K/AKT axis phosphorylates and inhibits the tuberous sclerosis complex (TSC1/2), thereby relieving its suppressive effect on mTORC1. Subsequently, activated mTORC1 phosphorylates downstream effectors such as S6K1 [142] and eukaryotic translation initiation factor 4E-BP1 [143], thereby facilitating protein synthesis and promoting cell proliferation. This process contributes to cell hypertrophy, a characteristic morphological feature of senescent cells [144] (Figure 4 ➀). The excessive accumulation of biomass under non-proliferative conditions is also involved in maintaining permanent cell cycle arrest, a key feature in the development of the senescence phenotype. mTOR further induces the expression of SASP factors, including pro-inflammatory cytokines and growth factors secreted by senescent cells. In particular, the translation of IL-1αis mTOR-dependent, and activation of mTOR signaling enhances IL-1α expression and promotes SASP induction (Figure 4 ➁). Moreover, mTORC1 negatively regulates autophagy through phosphorylation of the ULK1 complex (Unc-51-Like Autophagy-Activating Kinase 1, ATG13, FIP200 (FAK family kinase-interacting protein of 200 kDa), and ATG101), which is involved in the initiation of autophagy [105]. This phosphorylation-dependent inhibition directly suppresses autophagy initiation, supporting the role of mTORC1 as a negative regulator of autophagy (Figure 4 ➂). Therefore, excessive activation of mTORC1 impairs the proper clearance of misfolded proteins and damaged organelles, leading to the accumulation of intracellular stress, which in turn contributes to the progression of cellular senescence and an increased risk of age-related diseases. In contrast, under nutrient-deprived or starvation conditions, inhibition of mTORC1 allows activation of the ULK1 complex, thereby inducing autophagy. In senescent cells, however, mTORC1 activity remains persistently high, and sensitivity to nutrient and growth factor signals is diminished. This sustained mTORC1 activation is closely associated with the emergence of senescence-related phenotypes. Interestingly, even under conditions of extracellular nutrient depletion, intracellular amino acids produced by autophagy have been reported to maintain mTORC1 activity [145] (Figure 4 ➃), suggesting a potential positive feedback loop between autophagy and mTORC1. This dynamic interaction plays a central role in metabolic adaptation, maintenance of homeostasis, and regulation of senescence. In addition, mTOR functions as a key regulatory factor that determines cell fate—proliferation, senescence, or apoptosis—in response to external stimuli, such as cellular stress and nutrient availability, through interactions with p53 and AMPK. For instance, in response to stress stimuli such as DNA damage or energy depletion, p53 is known to suppress mTORC1 via activation of AMPK and induction of REDD1 expression [140] (Figure 4 ➄). This inhibition of mTORC1 leads to attenuation of cell growth and protein synthesis and promotes autophagy. Conversely, treatment with the mTORC1 inhibitor rapamycin reduces the binding of p53 mRNA to polysomes and decreases its translational efficiency, suggesting that mTORC1 directly regulates p53 protein synthesis [146] (Figure 4 ➅). Moreover, genotoxic stress, such as DNA damage, activates the mTOR-S6K1 signaling axis. The activated S6K1 exhibits increased affinity for MDM2, thereby attenuating MDM2-mediated ubiquitination of p53 and promoting its stabilization and accumulation [147] (Figure 4 ➆). Under specific conditions, such as in PTEN-deficient cells, both mTORC1 and mTORC2 have been reported to compete with MDM2 for binding to p53, promoting phosphorylation at Ser15 and increasing its transcriptional activity [148]. In this context, mTOR may positively regulate p53 stability and activation depending on the cellular conditions. Conversely, activation of the PI3K/AKT pathway induces phosphorylation of specific serine residues within MDM2, enhancing its nuclear translocation and protein stability. This post-translational modification augments MDM2’s E3 ubiquitin ligase activity toward p53, facilitating its proteasomal degradation [149] (Figure 4 ➇). Furthermore, mTORC1 activation has been shown to upregulate MDM2 translation, thereby reinforcing its role as a key negative regulator of p53 [150]. This regulatory mechanism is governed via the PI3K/AKT/mTOR axis. For instance, growth factors such as insulin-like growth factor 1 (IGF-1) and hepatocyte growth factor (HGF) activate the PI3K/AKT pathway, which in turn stimulates mTORC1. Activated mTORC1 enhances the translational efficiency of MDM2, resulting in elevated protein levels. The increased abundance of MDM2 promotes accelerated p53 degradation, thereby modulating cell proliferation and survival. This intricate regulatory interplay between the mTOR and p53 pathways under stress conditions has been elucidated in detail in prior investigations [32]. In summary, activation of the PI3K/AKT/mTOR pathway facilitates p53 degradation by promoting MDM2 translation (Figure 4 ➇).

Collectively, these findings demonstrate that the mTOR pathway exerts multifaceted effects on p53 expression and activity and plays a crucial role in determining cell fate. Conversely, p53 negatively regulates mTORC1 through AMPK activation and REDD1 induction, thereby maintaining cellular homeostasis by suppressing cell growth and survival [32]. Thus, mTOR acts as a bidirectional regulatory factor that modulates p53 activity positively or negatively depending on the cellular context through mechanisms involving translational control, regulation of protein stability, and post-translational modifications. The interaction between p53 and the mTOR pathway serves as a central control mechanism in the decision between cellular adaptation and proliferation or survival under stress stimuli. As shown in Figure 3 ➀, mTOR is also functionally connected to AMPK [103]. Together, p53, AMPK, and mTOR independently regulate autophagy, cell growth, senescence, and apoptosis while also forming an integrated regulatory network through close interactions. This network serves as a fundamental control system that enables cells to appropriately respond to environmental stress and make precise decisions regarding survival, repair, or cell death.

Autophagy is a tightly regulated intracellular catabolic mechanism that progresses through several distinct phases: initiation, nucleation, elongation and maturation, fusion, and ultimately degradation and recycling [151]. This subsection delineates the specific stages at which p53, AMPK, and mTORC1 exert regulatory influence. The process is triggered by the activation of the ULK1 complex, initiating the formation of phagophores. During the nucleation phase, the activated ULK1 complex stimulates the class III PI3K complex, promoting the generation of phosphatidylinositol 3-phosphate (PI3P) and thereby facilitating phagophore initiation. Subsequent expansion of the phagophore leads to the sequestration of cytoplasmic components. This stage involves ubiquitin-like conjugation systems that drive membrane elongation and closure, culminating in the formation of autophagosomes. Mature autophagosomes fuse with lysosomes to form autophagolysosomes, wherein the enclosed materials are enzymatically degraded. The resulting macromolecules are then recycled into the cytosol for reuse in cellular metabolism (Figure 5). Nuclear p53 functions as a transcriptional activator of several autophagy-related genes, including DRAM1, Sestrin1/2, TSC2, PTEN, and members of the ATG family, particularly under stress conditions such as genotoxic damage or nutrient scarcity [50,51,52,57]. Conversely, gain-of-function mutations in p53 have been shown to repress the transcription of key autophagy genes such as BECN1, DRAM1, and ATG12 while concurrently activating the mTOR pathway, thereby inhibiting autophagy and facilitating tumor cell survival and proliferation [152]. AMPK directly phosphorylates ULK1 at serine residues Ser317 and Ser777, activating the ULK1 complex and promoting autophagosome formation [105]. However, under nutrient-rich conditions, mTORC1 phosphorylates ULK1 at Ser757, thereby disrupting its interaction with AMPK and suppressing autophagy initiation [105].

#### 3.3.2. Functional Role of mTOR in Metabolism

The mTOR, similar to p53 and AMPK, plays a critical role in cellular energy homeostasis and metabolic regulation, with its activation closely associated with the metabolic status of the cell. In particular, mTORC1 serves as a major metabolic regulator that promotes cell growth and biosynthesis through the activation of glycolysis [153] (right panel of Figure 2). mTORC1 strongly enhances glycolytic flux via multiple mechanisms, including upregulation of the glucose transporter GLUT1, increased expression and activity of glycolytic enzymes, stimulation of hypoxia-inducible factor 1-alpha (HIF-1α) translation, and activation of the transcription factor MYC [153]. Specifically, mTORC1 promotes the localization of GLUT1 to the plasma membrane through the PI3K-Akt signaling pathway, thereby enhancing cellular glucose uptake. It also induces the expression of key glycolytic enzymes, including hexokinase 2 (HK2), PFKs, pyruvate kinase M2 (PKM2), and LDH, leading to increased glycolytic flux [154]. Additionally, mTORC1 facilitates the translation of HIF-1α, which in turn activates the transcription of glycolysis-related genes such as GLUT1, HK2, PKM2, and LDHA (lactate dehydrogenase A) [138,155]. Through the enhancement of MYC (MYC proto-oncogene, bHLH transcription factor) transcriptional activity, mTORC1 further augments the expression of glucose transporters and glycolytic enzymes, thereby promoting glucose metabolism [156]. Moreover, mTORC1 stimulates protein translation via phosphorylation of the 4E-BP, activating the translation of nuclear-encoded mitochondrial genes such as PGC-1α, TFAM (mitochondrial transcription factor A), YY1 (Yin Yang 1), mitochondrial ribosomal components, and subunits of the electron transport chain complexes I, II, IV, and V [140,157,158,159]. These actions enhance mitochondrial biogenesis and function, leading to the promotion of OXPHOS and increased ATP production capacity. Additionally, mTORC1 has been reported to regulate the transcription of glycolysis-related genes via NEAT1 (nuclear paraspeckle assembly transcript 1), a long non-coding RNA (lncRNA) [160]. This metabolic activation contributes to a reprogramming of the metabolic profile in senescent cells, suggesting the presence of a feedback loop between metabolism and senescence that can further modulate mTOR activity. Such a mechanism may be involved in the progression of cellular senescence and deterioration of metabolic function [118]. However, the glycolysis-promoting function of mTOR is known to be context-dependent. Under environmental conditions such as nutrient deprivation or oxidative stress, adaptive suppression of mTOR activity occurs to balance energy demand and survival strategies [161]. Thus, mTOR can be considered a central regulatory factor in metabolic sensing, plastically modulated in response to intra- and extracellular metabolic cues.

#### 3.3.3. Functional Role of mTOR in the Maintenance of Reproductive Function

Reduced autophagic activity has been reported to be closely associated with the progression of cellular senescence in various cell types, including hepatocytes [162], cardiomyocytes [163], and microglia [164]. For instance, in human cardiac progenitor cells, administration of the mTOR inhibitor rapamycin has been shown to alleviate replicative senescence and improve cellular function [165]. The mTOR signaling pathway functions as an aging-promoting factor in multiple model organisms, including Saccharomyces cerevisiae (budding yeast), Caenorhabditis elegans (nematodes), Drosophila melanogaster (fruit flies), and mice [46,166,167,168,169]. Moreover, genetic and pharmacological interventions such as overexpression of ATG genes, inhibition of mTOR signaling, or administration of autophagy-inducing compounds have been demonstrated to enhance autophagy, contributing to lifespan extension and delayed aging. In particular, inhibition strategies targeting both mTORC1 and mTORC2 complexes have garnered attention as potential therapeutic approaches for age regulation and age-related diseases, as they may enable reversible control of senescence-associated phenotypes in aging cells [169,170]. These findings underscore the protective role of autophagy in mitigating aging through the maintenance of cellular homeostasis and structural integrity. Dysregulation of mTOR signaling has been widely implicated in the onset and progression of various pathological conditions, including cancer, obesity, type 2 diabetes, metabolic syndrome, and neurodegenerative diseases such as Alzheimer’s disease [138,171,172]. Consequently, suppression of aging via mTOR inhibition remains an area of intense ongoing research.

In addition, mTOR has been shown to play a crucial regulatory role in maintaining reproductive function, including processes such as cellular senescence and decidualization. Baek et al. reported that, in human endometrial stromal cells, cAMP-induced decidualization is associated with increased activity of mTORC1 and decreased activity of mTORC2 [173]. In this study, mTORC1 activation was found to promote the expression of decidualization markers—prolactin (PRL) and insulin-like growth factor-binding protein 1 (IGFBP1)—through the transcription factor FOXO1, identifying mTORC1 as a key regulator of decidualization. Similarly, Hirota et al. demonstrated using a mouse model that hyperactivation of uterine mTORC1 signaling induces premature decidual senescence, resulting in preterm birth and fetal loss [13]. This pathological condition was shown to be ameliorated by low-dose administration of rapamycin, suggesting that proper regulation of mTORC1 signaling is essential for maintaining a normal pregnancy duration. Furthermore, a review by Guo et al. outlined the central role of mTOR signaling in female reproductive functions, including ovarian activity, cyclic endometrial remodeling, embryo implantation, and placental formation [14]. Notably, dysregulation of the mTOR pathway has been linked to gynecological disorders such as polycystic ovary syndrome (PCOS), premature ovarian failure (POF), and endometriosis. Collectively, these findings clearly demonstrate that mTOR signaling is indispensable for female reproductive function, particularly in the regulation of cellular senescence and decidualization.

### 3.4. Mechanisms of Cellular Stress Response and Senescence Regulation via Autophagy

This subsection outlines how nutritional status and cellular stress influence cellular senescence through the regulation of autophagy, mediated by the p53, AMPK, and mTOR signaling pathways. In cellular and microenvironmental conditions characterized by an abundance of glucose, amino acids, and growth factors such as insulin, the mTOR signaling pathway becomes activated. This activation promotes protein synthesis, glycolysis, and lipid synthesis and, consequently, enhances cell proliferation (Figure 6, left panel). Concurrently, AMPK is suppressed, and in the absence of cellular stress, p53 activity is also maintained at a low level. Under these conditions, autophagy is generally inhibited, which supports short-term cell growth by promoting anabolic metabolism and suppressing the catabolic process of autophagy. However, sustained inhibition of autophagy can lead to the accumulation of damaged organelles and misfolded proteins, which may impair cellular function over time [174]. Since autophagy plays a crucial role in the removal of these intracellular damaged components and in the prevention of senescence, prolonged autophagy inhibition in nutrient-rich conditions may contribute to the breakdown of cellular homeostasis and accelerate aging processes [53]. Therefore, while suppression of autophagy under nutrient-excess conditions may support cell proliferation in the short term, it could also promote cellular senescence in the long term through the accumulation of cellular damage.

Conversely, under nutrient-deprived conditions, cells halt proliferation and activate a metabolic program that prioritizes damage repair and energy conservation (Figure 6, right panel). In such states, mTOR activity is suppressed, leading to inhibition of protein synthesis and cell growth. Meanwhile, AMPK is activated, promoting fatty acid β-oxidation and the induction of autophagy via mTOR inhibition. Additionally, p53 is activated, inducing the expression of TIGAR, which suppresses glycolysis, and SCO2, which promotes mitochondrial oxidative phosphorylation. These changes, including cell cycle arrest and autophagy activation, facilitate the recycling of intracellular resources and the supply of essential metabolites, thereby contributing to cell survival and maintenance of function. Under specific conditions, autophagy induced through such metabolic adaptation has been shown to suppress the activation of senescence pathways. In other words, during acute and reversible stress, autophagy may function as a mechanism that counteracts senescence [175]. However, under chronic or excessive stress conditions, autophagy activation may instead promote cellular senescence [176]. In such aging processes, autophagy can provide the substrates necessary for the production of SASP factors, which act on the tissue microenvironment to reinforce and sustain the senescent state [177]. Furthermore, in response to stress signals such as oxidative stress, DNA damage, or oncogenic activation (e.g., Ras), p38 mitogen-activated protein kinase (p38 MAPK) is activated and has been shown to induce persistent senescence-associated phenotypes [178,179,180]. Activated p38 MAPK contributes to cellular senescence through the induction of cell cycle arrest and promotion of SASP. The involvement of p38 MAPK in senescence has been demonstrated in human endometrial-derived mesenchymal stem cells [181], human somatic stem cells [15], and fetal-derived cells [182].

There are at least two major types of cellular senescence. The first is telomere-dependent senescence, which is triggered by DNA damage responses caused by telomere shortening during cell division and involves activation of the p53/p21 pathway. The second is telomere-independent senescence, which is induced by growth factor stimuli such as insulin or oxidative stress, including damage from ROS, without telomere shortening [183]. In addition to these, factors such as oncogene activation, epigenetic modifications, and chronic inflammation also influence the regulation of cellular senescence. The effects of autophagy on senescence are context-dependent, varying with the nature of the stress, cellular differentiation stage or status, and the specific signaling pathways involved. Autophagy thus plays a dual role, capable of both suppressing and promoting senescence.

### 3.5. The Relationship Between Endometrial Cellular Senescence and Reproductive Capacity

Aging is a phenomenon characterized by the accumulation of widespread physiological changes over time, encompassing irreversible functional decline at the organismal level, including systemic deterioration of bodily functions, disruption of metabolic homeostasis, and reduced immune competence. In contrast, cellular senescence refers to a state in which cells undergo an irreversible arrest of proliferation, accompanied by distinctive morphological and functional changes. A hallmark of this state is the secretion of the SASP, which includes pro-inflammatory cytokines, chemokines, and growth factors. Cellular senescence is considered a contributing factor to aging, as the accumulation of senescent cells leads to impaired tissue function and chronic inflammation, thereby promoting the development of age-related diseases [184]. As illustrated in Figure 1, cellular senescence involves not only cell cycle arrest but also complex biological changes, including metabolic reprogramming. At the organismal level, age-related changes in metabolic profiles are also observed. For instance, in skeletal muscle, mitochondrial dysfunction leads to impaired OXPHOS, contributing to a decline in muscle mass and strength. Similarly, in endometrial cells, aging causes significant shifts in the balance between glycolysis and OXPHOS, leading to metabolic reprogramming. Specifically, aged endometrial cells exhibit hyperactivation of glycolysis [185], reduced OXPHOS, and mitochondrial dysfunction [186]. These metabolic abnormalities may impair decidualization and foster a pro-inflammatory microenvironment, ultimately diminishing endometrial receptivity and contributing to implantation failure, thereby negatively affecting reproductive function [187].

Several studies suggest that moderate cellular senescence may play a physiologically beneficial role in reproductive function, while excessive senescence could be associated with reduced implantation rates [13]. In fact, regulated cellular senescence plays an essential role in reproductive processes. For example, during the decidualization of endometrial stromal cells, the production of SASP factors is thought to enhance embryonic receptivity [188]. This controlled senescent state appears to be indispensable for establishing a receptive endometrial environment. However, excessive accumulation of senescent cells in the endometrium may exert detrimental effects on reproduction. According to a study by Tomari et al., human endometrial stromal cells from women with recurrent implantation failure exhibited elevated expression of the senescence markers p16 and p21, along with increased levels of SASP factors [189]. These changes were associated with decreased expression of stemness markers such as ABCG2 (ATP binding cassette subfamily G member 2) and ALDH1A1 (aldehyde dehydrogenase 1 family member A1), suggesting that excessive cellular senescence impairs endometrial regenerative capacity and reduces embryonic receptivity. Moreover, the disruption of interactions between senescent and immune cells is closely linked to implantation failure. Specifically, in women experiencing implantation failure, a functional disconnection has been observed between senescent endometrial cells and CD4^+^ T helper cells, indicating that impaired immune modulation may hinder successful implantation [190]. In summary, while moderate cellular senescence is essential for normal decidualization and the establishment of endometrial receptivity, excessive senescence may contribute to implantation failure through immune dysregulation and diminished regenerative potential.

Cellular senescence and aging are closely intertwined. In particular, key signaling pathways involved in the regulation of cellular senescence—such as p53, AMPK, and mTOR—serve as important modulators of age-related decline in reproductive function. Therapeutic strategies targeting these pathways to regulate cellular senescence may offer promising approaches for the treatment of age-related infertility.

## 4. Discussion

With advancing age, the functional capacity of the endometrium gradually declines, leading to a concomitant reduction in reproductive potential. This functional alteration is closely associated with cellular senescence in the endometrium. Endometrial cells are chronically exposed to cellular stressors such as DNA damage and oxidative stress, which force them into a binary fate—either entering an irreversible state of growth arrest known as senescence or maintaining survival. This process is primarily regulated by the DNA damage response pathway centered on p53 and is further modulated through crosstalk with AMPK and mTOR signaling pathways, which are involved in cellular energy metabolism and nutrient sensing [191]. In recent years, the complex interplay between cellular senescence and signaling pathways such as p53, AMPK, and mTOR has been actively investigated by many researchers. These molecules function as key regulators that coordinately control energy homeostasis, autophagy, and the progression of senescence, each via distinct mechanisms. Notably, their actions are context-dependent and often dualistic in nature.

For instance, under nutrient-deprived conditions, p53 and AMPK are activated, leading to the suppression of the mTOR pathway and a reduction in energy consumption. Conversely, in the presence of severe DNA damage or accumulated metabolic stress, persistent cell cycle arrest, induction of the SASP, and activation of the p38 MAPK pathway promote cellular senescence or apoptosis. On the other hand, if the cellular stress is mild and the damage is reversible, autophagy is induced, enabling the cell to escape senescence through transient cell cycle arrest [192]. Thus, p53, AMPK, and mTOR serve as central regulators that integrate cellular energy metabolism with the balance of growth, survival, senescence, and cell death. In addition to nutrient status, environmental factors such as chronic inflammation, oxidative stress involving epigenetic modifications, and DNA damage also play crucial roles in determining the cellular fate of endometrial cells—whether toward survival, senescence, or apoptosis.

Understanding the dualistic role of cellular senescence in the endometrium is essential for establishing therapeutic interventions targeting age-related reproductive decline. While appropriately regulated senescence responses contribute to optimizing endometrial receptivity, dysregulated senescence can impair endometrial function and reduce reproductive capacity. Recent attention has focused on the therapeutic potential of senolytics—agents such as quercetin and dasatinib—that selectively eliminate senescent cells. These compounds are expected to improve endometrial function and enhance implantation rates by removing dysfunctional senescent cells [193]. Findings from the present analysis suggest that p53 and AMPK activation, mTOR inhibition, the expression levels of local SASP factors, and the activation status of p38 MAPK in endometrial cells may serve as valuable biomarkers for assessing reproductive potential. For example, oxidative stress, exemplified by hydrogen peroxide exposure, activates the p38MAPK signaling pathway in human endometrial mesenchymal stem cells, thereby initiating an irreversible cell cycle arrest characteristic of cellular senescence [15]. Experimental evidence indicates that pharmacological inhibition of p38MAPK can partially reverse the senescent phenotype and reinstate proliferative capacity [15]. Age-related reproductive aging is believed to progress in a stepwise manner from a reversible to an irreversible pathological state; however, currently, no reliable biomarkers are available to distinguish these stages. Therefore, future research aimed at identifying and functionally characterizing such markers is anticipated to contribute to the prevention and treatment of reproductive aging.

The fundamental role of regulators such as p53, AMPK, and mTOR is thought to lie in their ability to suppress the proliferation of abnormal cells by inducing senescence, thereby functioning as a protective mechanism for tumor suppression [52]. At the same time, processes such as cell proliferation, differentiation, migration, tissue remodeling, and immune evasion, which are critical during cancer progression, also occur during embryogenesis—albeit under strict regulation. In embryogenesis, these processes are precisely controlled to ensure proper development and differentiation, whereas in cancer, the regulatory mechanisms are disrupted, leading to uncontrolled proliferation and tissue destruction. From this perspective, cancer can be regarded as a disease resulting from the aberrant reactivation of developmental regulatory pathways. Therefore, the fundamental difference between development and cancer may lie in the presence or absence of appropriate senescence control mechanisms. A deeper understanding of the similarities between embryogenesis and tumorigenesis may shed light on the molecular mechanisms underlying reproductive aging and aid in the development of novel therapeutic strategies. In particular, targeting signaling pathways specifically activated during early development may lead to innovative treatments capable of mitigating reproductive aging and preserving endometrial function. Continued elucidation of the mechanisms governing cellular senescence is expected to accelerate the development of effective interventions for age-related reproductive decline.

## 5. Conclusions

Signaling pathways such as p53, AMPK, and mTOR play pivotal roles in the regulation of endometrial cellular senescence and exert profound influences on reproductive function and the pathogenesis of associated disorders. A comprehensive understanding of the molecular crosstalk among these pathways holds promise for the development of novel therapeutic strategies aimed at preserving endometrial homeostasis and preventing or treating related pathologies. p53 acts as a key responder to various cellular stresses, including DNA damage and oxidative insults, and is instrumental in orchestrating cell cycle arrest, apoptosis, and the induction of cellular senescence. Within the endometrial context, p53 facilitates the suppression of mTORC1 signaling by activating AMPK, thereby modulating the progression of cellular senescence. AMPK, functioning as an intracellular energy sensor, is activated under conditions of ATP depletion and mitigates senescence by inhibiting cell proliferation and protein synthesis through the downregulation of mTORC1 activity. Conversely, mTOR represents a central regulatory axis that governs cellular growth and metabolic activity. Its hyperactivation has been implicated in the acceleration of cellular aging and oncogenic transformation. Activation of mTORC1 promotes anabolic processes, including cell growth and protein biosynthesis, yet its activity is stringently modulated by upstream regulators such as AMPK and p53. These interconnected pathways form a tightly regulated network that is essential for the maintenance of endometrial cellular integrity and homeostasis. Nevertheless, the functional roles of these molecular circuits are highly context-dependent, exhibiting pleiotropic effects that vary with environmental cues such as nutrient availability and oxidative stress. As a result, their regulatory functions are intricate and multifaceted, impacting a broad spectrum of cellular processes, including fate determination, metabolic control, and stress responses. Age-related progression of endometrial cellular senescence contributes significantly to the decline in reproductive capacity and the onset of infertility. Moreover, the accumulation of senescent cells is associated with an increased risk for pathological conditions such as endometriosis and endometrial carcinoma. Notably, dysregulation of p53 activity and aberrant mTOR signaling are critically implicated in abnormal cellular proliferation, tumorigenesis, and compromised reproductive outcomes, thereby making them attractive targets for therapeutic intervention. Looking forward, the integration of these molecular insights into clinical practice is anticipated to advance the realization of personalized medicine. By tailoring treatment strategies to the individual molecular profiles of patients, it will become feasible to implement precision therapies that optimize clinical efficacy while minimizing adverse effects.

## Figures and Tables

**Figure 1 cells-14-00858-f001:**
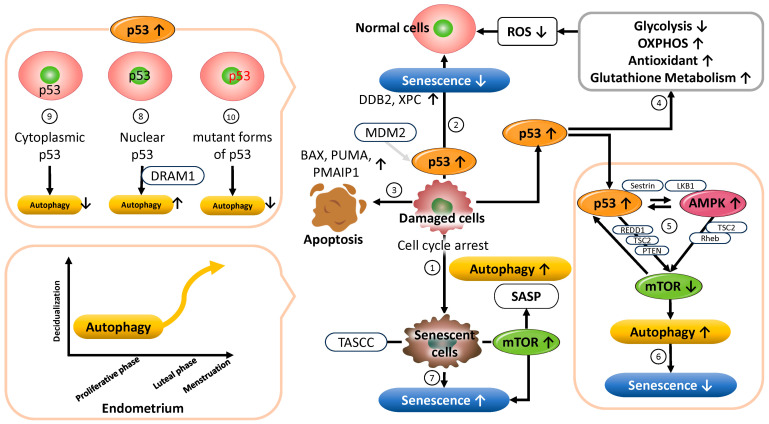
Functional role of p53 in cellular senescence and autophagy regulation. Stress activates p53, causing cell cycle arrest (➀); reversible damage avoids senescence (➁), while severe damage leads to apoptosis (➂). p53 also boosts OXPHOS and antioxidant defenses (➃), activates autophagy via AMPK and mTOR (➄,➅), but excess autophagy promotes SASP and senescence (➆). Localization affects p53’s role in autophagy (➇–➉). **Top-left bubble**: The subcellular localization of p53 critically modulates its regulatory function in autophagy (➇–➉). **Bottom-left bubble**: Autophagic activity in the endometrium exhibits dynamic fluctuations, increasing from the proliferative to the secretory phase, and is intricately linked to the process of decidualization. While some studies report a concomitant increase in both p53 expression and autophagy activity during the menstrual cycle, others present contrasting findings. Black arrows indicate facilitative or activating effects, whereas gray arrows denote inhibitory actions. AMPK, AMP-activated protein kinase; BAX, BCL2-associated X, apoptosis regulator; DDB2, damage-specific DNA-binding protein 2; DRAM1, DNA damage-regulated autophagy modulator 1; LKB1, liver kinase B1; mTOR, mechanistic target of rapamycin; OXPHOS, oxidative phosphorylation; PMAIP1, phorbol-12-myristate-13-acetate-induced protein 1; PTEN, phosphatase and tensin homolog; PUMA, p53 upregulated modulator of apoptosis; REDD1, regulated in development and DNA damage responses 1; Rheb, Ras homolog, mTORC1 binding; ROS, reactive oxygen species; SASP, senescence-associated secretory phenotype; TASCC, TOR-Autophagy Spatial Coupling Compartments; TSC2, TSC complex subunit 2; XPC, xeroderma pigmentosum group C.

**Figure 2 cells-14-00858-f002:**
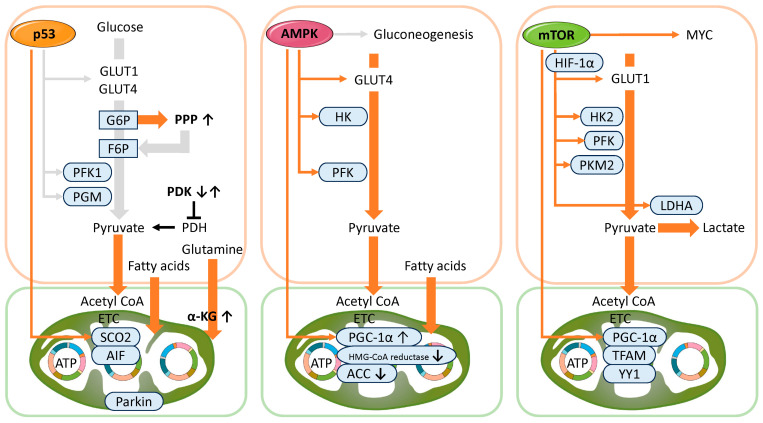
Functional role of p53 (**left**), AMPK (**middle**), and mTOR (**right**) in cellular metabolism. **Left panel**, p53 acts as a metabolic checkpoint by suppressing glycolysis through downregulation of glucose transporters and glycolytic enzymes. It promotes OXPHOS by inducing genes like SCO2 and enhances β-oxidation via activation of genes such as GAMT and Lipin1, thereby supporting efficient ATP production and reducing oxidative stress. **Middle panel**, AMPK stimulates glycolysis and β-oxidation to generate ATP and inhibits anabolic processes by suppressing mTOR activity, thus conserving energy and promoting catabolic pathways. **Right panel**, mTOR serves as a central regulator of cell growth and metabolism, promoting glycolysis and lipid synthesis under nutrient-rich conditions. However, its activation can inhibit autophagy and β-oxidation, potentially leading to energy imbalance and contributing to cellular senescence if not properly regulated. Together, these pathways coordinate to maintain energy balance, adapting cellular metabolism in response to varying energy demands and environmental conditions. Orange arrows indicate facilitative or activating effects, whereas gray arrows denote inhibitory actions. Orange boxes primarily denote glycolytic processes, whereas green boxes correspond to oxidative phosphorylation pathways. The colored rings depicted within the mitochondria represent mitochondrial DNA. ACC, acetyl-CoA carboxylase; AIF, apoptosis-inducing factor; AMPK, AMP-activated protein kinase; ATP, adenosine 5’ triphosphate; α-KG, α-ketoglutarate; ETC, electron transport chain; F6P, fructose-6-phosphate; G6P, glucose-6-phosphate; GLUT1, glucose transporter type 1; HIF-1α, hypoxia-inducible factor 1-alpha; HMG-CoA reductase, 3-hydroxy-3-methylglutaryl-CoA reductase; LDHA, lactate dehydrogenase A; MYC, MYC proto-oncogene, bHLH transcription factor; PDH, pyruvate dehydrogenase; PDK, pyruvate dehydrogenase kinase; PGC-1α, peroxisome proliferator-activated receptor gamma coactivator 1-alpha; PGM, phosphoglycerate mutase; PKM2, pyruvate kinase M2; PPP, pentose phosphate pathway; TFAM, mitochondrial transcription factor A; YY1, Yin Yang 1.

**Figure 3 cells-14-00858-f003:**
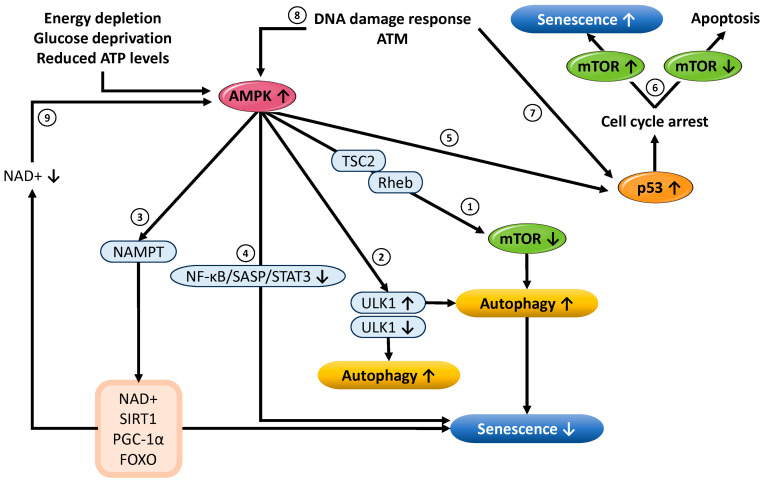
Functional role of AMPK in cellular senescence and autophagy regulation. AMPK regulates senescence via mTORC1 inhibition (➀) and autophagy through ULK1 (➁), though it may also suppress autophagy. It boosts NAD^+^ via NAMPT (➂), reducing stress and inflammation through NF-κB–SASP–STAT3 inhibition (➃). AMPK activates p53 under stress (➄), inducing p21 (➅). DNA damage activates ATM→p53 (➆), ATM→AMPK (➇), and NAD^+^/NADH shifts trigger AMPK→p53→p21 (➈). AMPK, AMP-activated protein kinase; FOXO, Forkhead box O; mTOR, mechanistic target of rapamycin; NAD+, nicotinamide adenine dinucleotide; NAMPT, nicotinamide phosphoribosyltransferase; NF-κB, Nuclear Factor kappa B; PGC-1α, peroxisome proliferator-activated receptor gamma coactivator 1-alpha; Rheb, Ras homolog, mTORC1 binding; SASP, senescence-associated secretory phenotype; SIRT1, sirtuin 1; STAT3, Signal Transducer and Activator of Transcription 3; TSC2, TSC complex subunit 2; ULK1, Unc-51-Like Autophagy-Activating Kinase 1.

**Figure 4 cells-14-00858-f004:**
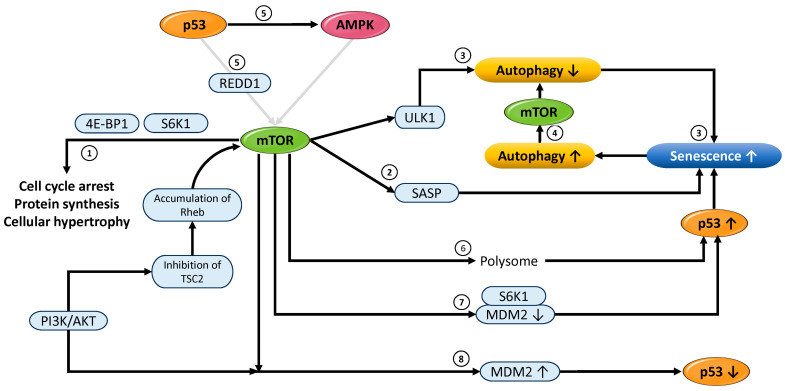
Functional role of mTOR in cellular senescence and autophagy regulation. mTORC1 promotes hypertrophy via S6K1/4E-BP1 (➀) and induces SASP through IL-1α translation (➁). It suppresses autophagy by ULK1 complex phosphorylation (➂). Even during starvation, autophagy-derived amino acids sustain mTORC1 (➃). p53-AMPK inhibits mTORC1 (➄), while mTORC1 regulates p53 via translation (➅), stabilization (➆), or degradation (➇). Black arrows indicate facilitative or activating effects, whereas gray arrows denote inhibitory actions. 4E-BP1, eIF4E-binding protein 1; AMPK, AMP-activated protein kinase; Akt, protein kinase B; MDM2, mouse double minute 2 homolog; mTOR, mechanistic target of rapamycin; PI3K, phosphoinositide 3-kinase; REDD1, regulated in development and DNA damage responses 1; S6K1, ribosomal S6 kinase; SASP, senescence-associated secretory phenotype; ULK1, Unc-51-Like Autophagy-Activating Kinase 1.

**Figure 5 cells-14-00858-f005:**
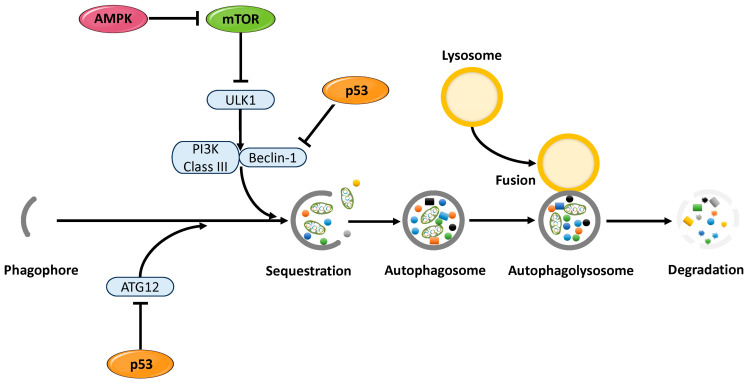
Regulatory involvement of p53, AMPK, and mTORC1 in the autophagy pathway. p53, AMPK, and mTORC1 primarily function at the initiation stage of autophagy. AMPK, AMP-activated protein kinase; ATG12, autophagy-related 12; mTOR, mechanistic target of rapamycin; PI3K, phosphoinositide 3-kinase; ULK1, Unc-51-Like Autophagy-Activating Kinase 1.

**Figure 6 cells-14-00858-f006:**
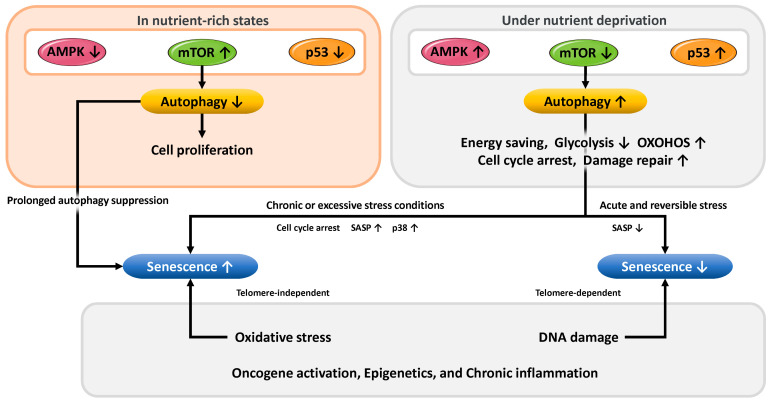
Mechanisms of cellular stress response and senescence regulation via autophagy. Nutrient-abundant conditions activate mTOR signaling while suppressing AMPK and p53 pathways, thereby inhibiting autophagy and promoting cellular growth. However, prolonged autophagy suppression may result in the accumulation of cellular damage, ultimately leading to the induction of senescence. Under nutrient deprivation, mTOR is suppressed, AMPK and p53 are activated, and autophagy is induced, aiding survival. Chronic stress or p38 MAPK activation promotes SASP and senescence. Autophagy has dual roles. AMPK, AMP-activated protein kinase; mTOR, mechanistic target of rapamycin; OXPHOS, oxidative phosphorylation; SASP, senescence-associated secretory phenotype.

## Data Availability

No new data were created.

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
