# Peer review of "Molecular Mechanisms of Cellular Senescence in Age-Related Endometrial Dysfunction"

_cells, 2025, doi:10.3390/cells14120858_

Round 1

Reviewer 1 Report

Comments and Suggestions for Authors

The manuscript entitled “Molecular Mechanisms of Cellular Senescence in Age-Related Endometrial Dysfunction” is well written and well-structured. The central theme of the paper is compelling and of significant scientific interest. However, I would like to offer a few suggestions for improvement:

  1. Minor revisions are needed to correct language inconsistencies throughout the manuscript.
  2. The inclusion of a figure illustrating the stages of autophagy is strongly recommended.
  3. A new figure depicting the PI3KCA/AKT/mTOR signaling pathway would enhance the clarity of the mechanistic discussion.
  4. Line 836 contains a typographical error: “...Regulation via AutophagyIn nu...” appears to be a merged phrase. Please revise accordingly.
  5. A table summarizing known inhibitors and activators of autophagy would be a valuable addition to the manuscript.
  6. The conclusion section should be expanded to provide a more comprehensive summary of the findings and their broader implications.
  7. The reference list should be updated to include more recent studies relevant to the field.

Author Response

Answer to the reviewers

Manuscript ID: cells-3679535

Type of manuscript: Review

Title: Molecular Mechanisms of Cellular Senescence in Age-Related Endometrial Dysfunction

Authors: Hiroshi Kobayashi *, Mai Umetani, Miki Nishio, Hiroshi Shigetomi, Shogo Imanaka, Hiratsugu Hashimoto

Dear Editor in Chief:

Cells

Thank you and the reviewers for the thoughtful comments and helpful suggestions on our manuscript. We have carefully considered each of the comments, made every effort to address the concerns raised, and applied corresponding revisions to the manuscript. The detailed, point-by-point responses to the reviewer comments are given below, whereas the corresponding revisions are highlighted to our manuscript within the document. The sentences in blue in the text were newly added.

We believe that our manuscript has been considerably improved as a result of these revisions, and hope that the revised manuscript is acceptable for publication in cells.

I would like to thank you once again for your consideration of my work and inviting me to submit the revised manuscript. I look forward to hearing from you.

Dear Editor,

While Reviewer 1 has indicated that the manuscript requires English editing, the other two reviewers have stated that the language is appropriate and does not require further revision. In light of this, I would like to confirm whether professional editing is still deemed necessary. I have prepared responses to Reviewer 1’s comments, and should they maintain the need for language editing, I am prepared to submit the manuscript for professional revision accordingly.

Additionally, Figure 1, bottom left: p53 expression across the menstrual cycle has been omitted from the figure due to inconsistencies in the reported data.

With best regards,

Hiroshi Kobayashi, M.D., Ph.D.

E-mail: hirokoba@naramed-u.ac.jp

Point-by-point responses to reviewer comments

Reviewer 1

Comment 1:

The manuscript entitled “Molecular Mechanisms of Cellular Senescence in Age-Related Endometrial Dysfunction” is well written and well-structured. The central theme of the paper is compelling and of significant scientific interest. However, I would like to offer a few suggestions for improvement:Minor revisions are needed to correct language inconsistencies throughout the manuscript.

Response 1:

The other two reviewers remarked that the English usage is appropriate and requires no further refinement. In this case, is it still necessary to proceed with professional proofreading? If you continue to believe it is warranted, please review the revised text below, and we will submit it for proofreading upon your approval.

Comment 2:

The inclusion of a figure illustrating the stages of autophagy is strongly recommended.

Response 2:

We were unable to identify definitive evidence indicating that p53, AMPK, or mTOR act at stages beyond the initiation of autophagy. Since schematic representations of the autophagic process are commonly included in standard textbooks, we initially deemed it unnecessary to incorporate stage-specific diagrams. However, in response to your valuable suggestion, we have added the following sentence to the concluding paragraph of Section 3.3.1, and created a new figure (Figure 5).

Autophagy is a tightly regulated intracellular catabolic mechanism that progresses through several distinct phases: initiation, nucleation, elongation and maturation, fusion, and ultimately degradation and recycling [151]. This subsection delineates the specific stages at which p53, AMPK, and mTORC1 exert regulatory influence. The process is triggered by the activation of the ULK1 complex, initiating the formation of phagophores. During the nucleation phase, the activated ULK1 complex stimulates the class III PI3K complex, promoting the generation of phosphatidylinositol 3-phosphate (PI3P) and thereby facilitating phagophore initiation. Subsequent expansion of the phagophore leads to the sequestration of cytoplasmic components. This stage involves ubiquitin-like conjugation systems that drive membrane elongation and closure, culminating in the formation of autophagosomes. Mature autophagosomes fuse with lysosomes to form autophagolysosomes, wherein the enclosed materials are enzymatically degraded. The resulting macromolecules are then recycled into the cytosol for reuse in cellular metabolism. Nuclear p53 functions as a transcriptional activator of several autophagy-related genes, including DRAM1, Sestrin1/2, TSC2, PTEN, and members of the ATG family, particularly under stress conditions such as genotoxic damage or nutrient scarcity [50–52, 57]. Conversely, gain-of-function mutations in p53 have been shown to repress the transcription of key autophagy genes such as BECN1, DRAM1, and ATG12, while concurrently activating the mTOR pathway, thereby inhibiting autophagy and facilitating tumor cell survival and proliferation [152]. AMPK directly phosphorylates ULK1 at serine residues Ser317 and Ser777, activating the ULK1 complex and promoting autophagosome formation [105]. However, under nutrient-rich conditions, mTORC1 phosphorylates ULK1 at Ser757, thereby disrupting its interaction with AMPK and suppressing autophagy initiation [105].

Figure 5. Regulatory involvement of p53, AMPK, and mTORC1 in the autophagy pathway.

p53, AMPK, and mTORC1 primarily function at the initiation stage of autophagy.

Comment 3:

A new figure depicting the PI3KCA/AKT/mTOR signaling pathway would enhance the clarity of the mechanistic discussion.

Response 3:

The PI3KCA/AKT/mTOR signaling axis has been comprehensively characterized by Pavlova et al. [Pavlova JA]. The PI3KCA/AKT/TSC/mTOR/MDM2 signaling cascade has been incorporated into Figure 4, accompanied by an explanatory addition to Section 3.3.1.

The PI3K/AKT/mTOR pathway is a pivotal intracellular signaling cascade that orchestrates a wide spectrum of cellular functions, including growth, proliferation, metabolism, and survival. Upon activation by growth factors and other extracellular cues, the PI3K/AKT axis phosphorylates and inhibits the tuberous sclerosis complex (TSC1/2), thereby relieving its suppressive effect on mTORC1. Subsequently, activated mTORC1 phosphorylates downstream effectors such as S6K1 [142] and eukaryotic translation initiation factor 4E-BP1 [143], thereby facilitating protein synthesis and promoting cell proliferation. This process contributes to cell hypertrophy, a characteristic morphological feature of senescent cells [144] (Figure 4①).

Moreover, genotoxic stress, such as DNA damage, activates the mTOR-S6K1 signaling axis. The activated S6K1 exhibits increased affinity for MDM2, thereby attenuating MDM2-mediated ubiquitination of p53 and promoting its stabilization and accumulation [147] (Figure 4⑦).

Conversely, activation of the PI3K/AKT pathway induces phosphorylation of specific serine residues within MDM2, enhancing its nuclear translocation and protein stability. This post-translational modification augments MDM2's E3 ubiquitin ligase activity toward p53, facilitating its proteasomal degradation [149] (Figure 4⑧). Furthermore, mTORC1 activation has been shown to upregulate MDM2 translation, thereby reinforcing its role as a key negative regulator of p53 [150]. This regulatory mechanism is governed via the PI3K/AKT/mTOR axis. For instance, growth factors such as insulin-like growth factor 1 (IGF-1) and hepatocyte growth factor (HGF) activate the PI3K/AKT pathway, which in turn stimulates mTORC1. Activated mTORC1 enhances the translational efficiency of MDM2, resulting in elevated protein levels. The increased abundance of MDM2 promotes accelerated p53 degradation, thereby modulating cell proliferation and survival. This intricate regulatory interplay between the mTOR and p53 pathways under stress conditions has been elucidated in detail in prior investigations [32]. In summary, activation of the PI3K/AKT/mTOR pathway facilitates p53 degradation by promoting MDM2 translation (Figure 4⑧).

Comment 4:

Line 836 contains a typographical error: “...Regulation via AutophagyIn nu...” appears to be a merged phrase. Please revise accordingly.

Response 4:

In the submitted manuscript, the text appears as follows, suggesting that the issue may have arisen during the conversion to Word format.

Figure 6. Mechanisms of Cellular Stress Response and Senescence Regulation via Autophagy

In nutrient-rich states, mTOR is activated, AMPK and p53 are suppressed, and autophagy is inhibited, promoting

Comment 5:

A table summarizing known inhibitors and activators of autophagy would be a valuable addition to the manuscript.

Response 5:

The overview and mechanisms of action of major autophagy inhibitors and activators have been comprehensively delineated in previous studies [Yang YP; Shi Q; Pavlova JA]. A representative list of these pharmacological agents is presented in the figure below.

Yang YP, Hu LF, Zheng HF, Mao CJ, Hu WD, Xiong KP, Wang F, Liu CF. Application and interpretation of current autophagy inhibitors and activators. Acta Pharmacol Sin. 2013 May;34(5):625-35. doi: 10.1038/aps.2013.5.

Shi Q, Pei F, Silverman GA, Pak SC, Perlmutter DH, Liu B, Bahar I. Mechanisms of Action of Autophagy Modulators Dissected by Quantitative Systems Pharmacology Analysis. Int J Mol Sci. 2020 Apr 19;21(8):2855. doi: 10.3390/ijms21082855.

Pavlova JA, Guseva EA, Dontsova OA, Sergiev PV. Natural Activators of Autophagy. Biochemistry (Mosc). 2024 Jan;89(1):1-26. doi: 10.1134/S0006297924010012.

Autophagy Modulators Overview

Activators of Autophagy

Activator

Mechanism of Action

Notes

Rapamycin

Inhibits mTORC1, leading to autophagy induction

Widely used in research and clinical settings

Metformin

Activates AMPK, which inhibits mTOR and promotes autophagy

Common anti-diabetic drug with autophagy-inducing properties

Resveratrol

Activates SIRT1 and AMPK pathways

Natural polyphenol found in grapes and red wine

Spermidine

Induces autophagy via epigenetic regulation and acetyltransferase inhibition

Naturally occurring polyamine with anti-aging effects

Urolithin A

Promotes mitophagy through PINK1/Parkin pathway activation

Metabolite derived from ellagitannins found in pomegranates

AICAR

Activates AMPK, leading to autophagy induction

AMP analog that mimics energy stress

Inhibitors of Autophagy

Inhibitor

Mechanism of Action

Notes

Chloroquine

Inhibits lysosomal acidification, blocking autophagosome-lysosome fusion

Used in malaria treatment; repurposed for cancer therapy

Bafilomycin A1

Inhibits vacuolar H⁺-ATPase, preventing lysosome acidification

Potent autophagy inhibitor in research

3-Methyladenine

Inhibits class III PI3K, blocking autophagosome formation

Commonly used in laboratory studies

Wortmannin

Inhibits PI3K, affecting early stages of autophagy

Broad-spectrum kinase inhibitor

Spautin-1

Promotes degradation of Beclin 1 by inhibiting USP10/13 deubiquitinases

Experimental compound targeting autophagy initiation

Numerous studies have reported pharmacological interventions aimed at attenuating endometrial cell senescence. These compounds—classified as senescence inhibitors—exert secondary effects on autophagy but are not direct autophagy modulators (i.e., inhibitors or activators) (see subsequent text). To date, no clinical evidence supports the notion that autophagy modulators can ameliorate endometrial cell senescence or restore reproductive competence in humans. Importantly, these compounds—classified as senolytics—exert secondary effects on autophagy but are not direct autophagy modulators (i.e., inhibitors or activators). Given that the primary objective of this review is to elucidate the molecular mechanisms underlying age-associated endometrial senescence, we would like to omit the presentation of known autophagy inhibitors and activators used to explain the molecular mechanism, as this may confuse readers.

The following is a synopsis of studies examining pharmacologic agents that mitigate endometrial cellular senescence. Inhibition of autophagy may exacerbate endometrial aging by disrupting cellular homeostasis. For instance, pharmacological blockade of autophagy using agents such as chloroquine (CQ) has been demonstrated to induce epithelial-mesenchymal transition (EMT) in endometrial epithelial cells, thereby contributing to fibrotic remodeling—a hallmark of intrauterine adhesions (IUA). This effect is mechanistically linked to downregulation of DIO2 and activation of the MAPK/ERK-mTOR signaling cascade (PMID: 35196191). Furthermore, autophagy inhibition attenuates the expression of decidualization markers, including HOXA10 and the progesterone receptor (PR), thereby impairing decidual transformation and negatively impacting embryo implantation and early pregnancy outcomes (PMID: 32072231; 31293650). Conversely, pharmacologic activation of autophagy appears to counteract several aspects of endometrial aging. Agents such as rapamycin have been shown to restore EMT and mitigate fibrosis in endometrial tissues affected by IUA, effects mediated via restoration of DIO2 expression and modulation of the MAPK/ERK-mTOR pathway (PMID: 35196191). In women with polycystic ovary syndrome (PCOS), diminished expression of autophagy-related genes correlates with endometrial dysfunction and elevated androgen levels; this dysregulation can be partially reversed by metformin (PMID: 27845161). Moreover, autophagy activators promote decidualization in endometrial stromal cells. For example, rapamycin has been reported to restore autophagic flux through the AMPK/mTOR pathway, thereby rescuing decidualization compromised by folate deficiency (PMID: 31234569; 33877643). Additionally, estrogen has been found to inhibit autophagy in endometrial stromal cells, while progesterone may counteract this effect and enhance autophagic activity. The hormonal balance between estrogen and progesterone is thus essential to sustain physiologic autophagy throughout the menstrual cycle (PMID: 31293650).

Key studies addressing pharmacological suppression of endometrial senescence include:

Quercetin and Dasatinib as Senolytic Agents in Endometrial Stromal Cells

Quercetin and dasatinib, both classified as senolytics, selectively eliminate senescent cells. One study demonstrated that these agents suppress senescence in endometrial stromal cells (ESCs) while promoting decidualization. Specifically, quercetin reduced expression of senescence markers and upregulated decidualization markers such as IGFBP1, PRL, and FOXO1. Moreover, the combination of quercetin and dasatinib exhibited synergistic effects superior to monotherapy [doi: 10.1016/j.bbrc.2021.07.075]. While both agents ultimately influence autophagic pathways, they are not direct autophagy modulators.Quercetin Enhances Decidualization via the AKT–ERK–p53 Pathway

This study revealed that quercetin inhibits phosphorylation of AKT, ERK1/2, and PRAS40, thereby suppressing production of senescence-associated secretory phenotype (SASP) factors such as IL-6 and MMP3. Concurrently, quercetin increased both the phosphorylation and total protein levels of p53, promoting apoptosis in senescent cells and enhancing decidualization [doi: 10.1186/s12958-024-01265-z].

Megestrol Acetate Induces Senescence in Endometrial Cancer Cells

Megestrol acetate has been reported to induce senescence in endometrial cancer cells via upregulation of p21 and p16 through progesterone receptor B (PR-B) signaling. This effect is mediated by the FOXO1 pathway, as FOXO1 inhibition attenuates megestrol-induced senescence. Although megestrol acetate influences autophagy indirectly, it is not classified as an autophagy inhibitor or activator.

Collectively, these findings highlight the potential of pharmacological agents in modulating endometrial cell senescence, offering promising avenues for the treatment of infertility and endometriosis. Notably, senolytic agents such as quercetin and dasatinib may enhance endometrial function by facilitating the removal of senescent cells. Future research should prioritize evaluating the clinical safety and efficacy of these compounds. Furthermore, a deeper understanding of the molecular underpinnings of endometrial senescence is anticipated to inform the development of more targeted therapeutic strategies. In conclusion, maintenance of optimal autophagy is crucial for endometrial integrity. Excessive suppression of autophagy may precipitate fibrotic changes and impair reproductive outcomes, whereas judicious activation of autophagic pathways can mitigate senescence-associated dysfunction. While therapeutic modulation of autophagy represents a compelling strategy, current clinical evidence does not support its efficacy in reversing cellular senescence or restoring reproductive capacity in the human endometrium.

Once again, numerous studies have reported pharmacological interventions aimed at attenuating endometrial cell senescence. However, these compounds—classified as senolytics—exert secondary effects on autophagy but are not direct autophagy modulators (i.e., inhibitors or activators). Senolytic agents such as quercetin and dasatinib exert their effects through multiple mechanisms beyond the modulation of autophagy. While they can influence autophagic pathways, their primary actions involve the selective elimination of senescent cells by targeting specific pro-survival signaling networks. Dasatinib, a tyrosine kinase inhibitor, induces apoptosis in senescent cells by inhibiting Src family kinases. Quercetin targets anti-apoptotic proteins such as BCL-xL and HIF-1α, disrupting the survival of senescent endothelial cells and promoting their clearance. Additionally, the combination of dasatinib and quercetin (D+Q) has been shown to activate autophagy, contributing to the alleviation of cellular dedifferentiation in certain contexts, such as diabetic kidney disease. Therefore, while autophagy modulation is one aspect of their activity, the senolytic effects of quercetin and dasatinib primarily arise from their ability to disrupt key survival pathways in senescent cells, leading to apoptosis and improved tissue function.

Comment 6:

The conclusion section should be expanded to provide a more comprehensive summary of the findings and their broader implications.

Response 6:

The conclusion section expands on the discussion as follows:

Signaling pathways such as p53, AMPK, and mTOR play pivotal roles in the regulation of endometrial cellular senescence and exert profound influences on reproductive function and the pathogenesis of associated disorders. A comprehensive understanding of the molecular crosstalk among these pathways holds promise for the development of novel therapeutic strategies aimed at preserving endometrial homeostasis and preventing or treating related pathologies. p53 acts as a key responder to various cellular stresses, including DNA damage and oxidative insults, and is instrumental in orchestrating cell cycle arrest, apoptosis, and the induction of cellular senescence. Within the endometrial context, p53 facilitates the suppression of mTORC1 signaling by activating AMPK, thereby modulating the progression of cellular senescence. AMPK, functioning as an intracellular energy sensor, is activated under conditions of ATP depletion and mitigates senescence by inhibiting cell proliferation and protein synthesis through the downregulation of mTORC1 activity. Conversely, mTOR represents a central regulatory axis that governs cellular growth and metabolic activity. Its hyperactivation has been implicated in the acceleration of cellular aging and oncogenic transformation. Activation of mTORC1 promotes anabolic processes, including cell growth and protein biosynthesis, yet its activity is stringently modulated by upstream regulators such as AMPK and p53. These interconnected pathways form a tightly regulated network that is essential for the maintenance of endometrial cellular integrity and homeostasis. Nevertheless, the functional roles of these molecular circuits are highly context-dependent, exhibiting pleiotropic effects that vary with environmental cues such as nutrient availability and oxidative stress. As a result, their regulatory functions are intricate and multifaceted, impacting a broad spectrum of cellular processes including fate determination, metabolic control, and stress responses. Age-related progression of endometrial cellular senescence contributes significantly to the decline in reproductive capacity and the onset of infertility. Moreover, the accumulation of senescent cells is associated with an increased risk for pathological conditions such as endometriosis and endometrial carcinoma. Notably, dysregulation of p53 activity and aberrant mTOR signaling are critically implicated in abnormal cellular proliferation, tumorigenesis, and compromised reproductive outcomes, thereby making them attractive targets for therapeutic intervention. Looking forward, the integration of these molecular insights into clinical practice is anticipated to advance the realization of personalized medicine. By tailoring treatment strategies to the individual molecular profiles of patients, it will become feasible to implement precision therapies that optimize clinical efficacy while minimizing adverse effects.

Comment 7:

The reference list should be updated to include more recent studies relevant to the field.

Response 7:

We conducted an updated literature search spanning the period from 2024 to the present. Two previously cited publications (PMID: 39118090, PMID: 39487595) were identified that addressed the keywords “cellular senescence,” “endometrium,” and “p53.” Similarly, two additional studies (PMID: 27941214, PMID: 27454290) focusing on “cellular senescence,” “endometrium,” and “AMPK” were also retrieved and have already been cited in this review. One newly identified publication (PMID: 38770820) explored the association between “cellular senescence,” “endometrium,” and “mTOR.” This study demonstrated that a combinatorial treatment involving human umbilical cord-derived mesenchymal stem cells—known for their regenerative and reparative capacities—and the antioxidant dehydroepiandrosterone (DHEA) effectively attenuated senescence in murine endometrial epithelial cells induced by oxidative stress and inflammation. This study was not cited as its relevance to the central theme of this review is limited.

A total of six articles pertaining to “cellular senescence,” “endometrium,” and the signaling molecules “p53,” “AMPK,” and “mTOR” were examined:

DOI: 10.3389/fonc.2024.1455492

This review comprehensively assessed the mechanistic role of metformin in cancer therapy, with a particular emphasis on glioblastoma. The study emphasized the context-dependent effects of metformin, highlighting that cell fate decisions—apoptosis versus senescence—are influenced by cancer cell type, PTEN and p53 mutation status, and mTOR activity. This paper has been cited in Section 3.1.1.

DOI: 10.1186/s40001-025-02562-y

This study revealed that quercetin mitigates oxidized LDL-induced senescence in aortic endothelial cells and macrophages through modulation of the p16/p21, p53/SERPINE1, and AMPK/mTOR pathways, offering a novel anti-atherogenic mechanism. Due to its lack of direct relevance to endometrial physiology, this paper was not cited.

DOI: 10.1016/j.jot.2024.03.001

The authors demonstrated that metformin ameliorates rapid mandibular bone loss in a mouse model of premature aging by correcting dysregulated molecular pathways related to cellular energy sensing, redox balance, senescence, and osteoclastogenesis. This article was not cited due to its limited applicability to the review’s focus.

DOI: 10.3390/md22030127

Extracts from Corydalis heterocarpa (CHE), a halophytic plant, were shown to possess various bioactivities, including anti-inflammatory, anti-cancer, and anti-adipogenic effects. CHE markedly downregulated aging-related regulators such as p53 and p21, suggesting potential anti-aging properties. This study was excluded due to insufficient relevance.

DOI: 10.3390/antiox13070759

This investigation proposed that delphinidin (Delp), a natural antioxidant, exerts protective effects against oxidative stress in human nucleus pulposus cells by enhancing autophagy via the SIRT1/AMPK/mTOR signaling axis. Given its focus on intervertebral disc degeneration rather than endometrial senescence, it was not cited.

Reviewer 2 Report

Comments and Suggestions for Authors

The manuscript provides a comprehensive review of the molecular mechanisms underlying cellular senescence in the endometrium, focusing on the roles of the p53, AMPK, and mTOR pathways. This topic is highly relevant to reproductive aging and endometrial dysfunction. The manuscript is well-written and presents valuable findings. However, I have a few minor suggestions that could help improve clarity and presentation:

  1. The term "inflammaging" is introduced but not thoroughly defined or linked to endometrial senescence. A brief explanation of how chronic inflammation contributes to endometrial dysfunction would enhance the discussion.
  2. The figures (e.g., Figure 1 and Figure 2) are informative but could benefit from additional labeling or legends to guide readers. For instance, Figure 1 includes numbered pathways (①–⑩), but the accompanying text does not explicitly reference these numbers, making it harder to follow.
  3. The manuscript mentions biomarkers (e.g., p38 MAPK activation) for assessing reproductive potential. Could these biomarkers be translated into diagnostic tools? A brief discussion on this topic would be useful.

Author Response

Answer to the reviewers

Manuscript ID: cells-3679535

Type of manuscript: Review

Title: Molecular Mechanisms of Cellular Senescence in Age-Related Endometrial Dysfunction

Authors: Hiroshi Kobayashi *, Mai Umetani, Miki Nishio, Hiroshi Shigetomi, Shogo Imanaka, Hiratsugu Hashimoto

Dear Editor in Chief:

Cells

Thank you and the reviewers for the thoughtful comments and helpful suggestions on our manuscript. We have carefully considered each of the comments, made every effort to address the concerns raised, and applied corresponding revisions to the manuscript. The detailed, point-by-point responses to the reviewer comments are given below, whereas the corresponding revisions are highlighted to our manuscript within the document. The sentences in blue in the text were newly added.

We believe that our manuscript has been considerably improved as a result of these revisions, and hope that the revised manuscript is acceptable for publication in cells.

I would like to thank you once again for your consideration of my work and inviting me to submit the revised manuscript. I look forward to hearing from you.

Dear Editor,

While Reviewer 1 has indicated that the manuscript requires English editing, the other two reviewers have stated that the language is appropriate and does not require further revision. In light of this, I would like to confirm whether professional editing is still deemed necessary. I have prepared responses to Reviewer 1’s comments, and should they maintain the need for language editing, I am prepared to submit the manuscript for professional revision accordingly.

Additionally, Figure 1, bottom left: p53 expression across the menstrual cycle has been omitted from the figure due to inconsistencies in the reported data.

With best regards,

Hiroshi Kobayashi, M.D., Ph.D.

E-mail: hirokoba@naramed-u.ac.jp

Point-by-point responses to reviewer comments

Reviewer 2

Comment 1:

The manuscript provides a comprehensive review of the molecular mechanisms underlying cellular senescence in the endometrium, focusing on the roles of the p53, AMPK, and mTOR pathways. This topic is highly relevant to reproductive aging and endometrial dysfunction. The manuscript is well-written and presents valuable findings. However, I have a few minor suggestions that could help improve clarity and presentation:

The term "inflammaging" is introduced but not thoroughly defined or linked to endometrial senescence. A brief explanation of how chronic inflammation contributes to endometrial dysfunction would enhance the discussion.

Response 1:

The following sentence has been added to the first paragraph of Section 3:

I Inflammaging, characterized by chronic, low-grade inflammation linked to aging, plays a pivotal role in promoting endometrial senescence, thereby contributing to endometrial dysfunction [20]. This process involves the secretion of proinflammatory mediators collectively known as the SASP, and the resulting sustained inflammatory milieu disrupts tissue homeostasis and diminishes endometrial receptivity, ultimately impairing fertility and pregnancy outcomes [20].

Comment 2:

The figures (e.g., Figure 1 and Figure 2) are informative but could benefit from additional labeling or legends to guide readers. For instance, Figure 1 includes numbered pathways (①–⑩), but the accompanying text does not explicitly reference these numbers, making it harder to follow.

Response 2:

Items ① to ⑩ are clearly stated in the main text of 3.1.1.

If they are not displayed, we are concerned that there may be a problem with the character conversion software used to download the file.

Comment 3:

The manuscript mentions biomarkers (e.g., p38 MAPK activation) for assessing reproductive potential. Could these biomarkers be translated into diagnostic tools? A brief discussion on this topic would be useful.

Response 3:

The following sentence has been added to the third paragraph of the Discussion:

For example, oxidative stress, exemplified by hydrogen peroxide exposure, activates the p38MAPK signaling pathway in human endometrial mesenchymal stem cells, thereby initiating an irreversible cell cycle arrest characteristic of cellular senescence [15]. Experimental evidence indicates that pharmacological inhibition of p38MAPK can partially reverse the senescent phenotype and reinstate proliferative capacity [15].

Reviewer 3 Report

Comments and Suggestions for Authors

Comments to the Authors,

The manuscript titled “Molecular Mechanisms of Cellular Senescence in Age-Related Endometrial Dysfunction” (Manuscript number: cells-3679535), presents an innovative high-quality approach in which established bibliometric tools were utilized to provide in-depth information on the role of cellular senescence in age-related endometrial dysfunction and the underlying molecular mechanisms. The topic is of great interest and the manuscript presents a plurality of published studies with data aiming to provide an overall insight toward the influence of cellular senescence in endometrial functionality as well as the future directions for potential therapeutic options regarding different pathological conditions including infertility, endometriosis and endometrial carcinoma.

Author Response

Answer to the reviewers

Manuscript ID: cells-3679535

Type of manuscript: Review

Title: Molecular Mechanisms of Cellular Senescence in Age-Related Endometrial Dysfunction

Authors: Hiroshi Kobayashi *, Mai Umetani, Miki Nishio, Hiroshi Shigetomi, Shogo Imanaka, Hiratsugu Hashimoto

Dear Editor in Chief:

Cells

Thank you and the reviewers for the thoughtful comments and helpful suggestions on our manuscript. We have carefully considered each of the comments, made every effort to address the concerns raised, and applied corresponding revisions to the manuscript. The detailed, point-by-point responses to the reviewer comments are given below, whereas the corresponding revisions are highlighted to our manuscript within the document. The sentences in blue in the text were newly added.

We believe that our manuscript has been considerably improved as a result of these revisions, and hope that the revised manuscript is acceptable for publication in cells.

I would like to thank you once again for your consideration of my work and inviting me to submit the revised manuscript. I look forward to hearing from you.

Dear Editor,

While Reviewer 1 has indicated that the manuscript requires English editing, the other two reviewers have stated that the language is appropriate and does not require further revision. In light of this, I would like to confirm whether professional editing is still deemed necessary. I have prepared responses to Reviewer 1’s comments, and should they maintain the need for language editing, I am prepared to submit the manuscript for professional revision accordingly.

Additionally, Figure 1, bottom left: p53 expression across the menstrual cycle has been omitted from the figure due to inconsistencies in the reported data.

With best regards,

Hiroshi Kobayashi, M.D., Ph.D.

E-mail: hirokoba@naramed-u.ac.jp

Point-by-point responses to reviewer comments

Reviewer 3

Comment 1:

The manuscript titled “Molecular Mechanisms of Cellular Senescence in Age-Related Endometrial Dysfunction” (Manuscript number: cells-3679535), presents an innovative high-quality approach in which established bibliometric tools were utilized to provide in-depth information on the role of cellular senescence in age-related endometrial dysfunction and the underlying molecular mechanisms. The topic is of great interest and the manuscript presents a plurality of published studies with data aiming to provide an overall insight toward the influence of cellular senescence in endometrial functionality as well as the future directions for potential therapeutic options regarding different pathological conditions including infertility, endometriosis and endometrial carcinoma.

Response 1:

We thank you for your careful review.